



# Methane in the Danube Delta: The importance of spatial patterns and diel cycles for atmospheric emission estimates

Anna Canning[1], Bernhard Wehrli[2,3], and Arne Körtzinger[1,4]

[1]GEOMAR Helmholtz-Zentrum für Ozeanforschung, Kiel, Schleswig-Holstein, Germany
[2]Institute of Biogeochemistry and Pollutant Dynamics, ETH Zürich, Zürich, 8092, Switzerland
[3]Eawag, Swiss Federal Institute of Aquatic Science and Technology, Kastanienbaum, 6047, Switzerland
[4]Christian-Albrechts-Universität zu Kiel, Kiel, Schleswig-Holstein, Germany

**Correspondence:** Anna Canning (acanning@geomar.de)

**Abstract.**

Methane ($CH_4$) is one of the substantial greenhouse gases in our atmosphere and its concentration has increased by $\sim 4\%$ over the last decade. Although sources driving these increases are not well constrained, one potential contribution comes from wetlands, which are usually intertwined with rivers, channels and lakes, creating a considerable need to acquire higher resolu-

tion data to facilitate modelling and predictions. Here we took a fully contained sensor set-up to obtain measurements of $CO_2$, $CH_4$, $O_2$ and auxiliary parameters, installed on a houseboat for accessibility, to assess and analyse surface water concentrations within the Danube Delta, Romania. Over 3 seasons, we transected a $\sim 400$ km route with concentration mapping and additional stations for monitoring diel cycles. Overall, the delta was a source for $CH_4$ throughout all seasons, with concentrations ranging between $0.113$–$15.6$ $\mu$mol $L^{-1}$. The dataset was split into three different subsystems; lakes, rivers and channels, with channels

showing the highest variability. We found large to extreme diel cycles in both the lakes and channels, with concentrations varying by an order of magnitude between these two systems. The observed strong diel cycle within the lake suggests daily vertical stratification allowing for macrophytes to create a temporal oxycline due to lack of light and movement between the stems as previously suggested. While throughout the day, there was a consistent overall surface concentration of $CH_4$ at around $0.4$ $\mu$mol $L^{-1}$, there was a clear linear trend with an $O_2$:$CH_4$ molar ratio of -50:1 during the phase of nocturnal convection

with the two water stratified bodies mixing during the night. Daily spot sampling techniques and neglecting such diel cycles reducing the estimated average methane concentrations by $25\%$ and increase by $3.3\%$ for channels and lakes, respectively. On an individual lake basis, spot sampling can potentially incur an uncertainty range of a factor of 4.5. Analyses also included a 'hot spot', with a 10-fold stronger methane increase ($4$–$15.6$ $\mu$mol $L^{-1}$) overnight compared to the lake, with an almost immediate and extreme decrease in $CH_4$ following sunrise. Calculated diffusive $CH_4$ fluxes for the overall delta yielded an

average of $49 \pm 61$ $\mu$mol $m^{-2}$ $h^{-1}$ corresponding to an extrapolated annual flux of $0.43 \pm 0.53$ mol $m^{-2}$ $yr^{-1}$. Our data illustrate the importance of collecting information on diel cycles in different habitats to improve the emission estimates from wetland systems.



## 1 Introduction

Methane ($CH_4$) is one of the most relevant anthropogenic greenhouse gases following carbon dioxide ($CO_2$) with an estimated
global emission rate of 572 Tg $CH_4$ $yr^{-1}$ for the decade 2003-2012 (Saunois et al. 2019). More recently, we have seen an accelerated increase from 1775 ppb in 2006 to 1850 ppb in 2017, and over a 100-year interval, $CH_4$ is 34 times more potent as a greenhouse gas than $CO_2$ when including climate carbon feedbacks (28 times without feedbacks: Myhre et al. 2013; Schubert and Wehrli 2019), and its continued increase has the potential to reverse any progress made for climate mitigation by reducing $CO_2$ emissions (Nisbet et al. 2019). Biogenic emissions from wetlands are a potential driver (Nisbet et al. 2019),
contributing to the overall estimate of 159 (117–212) Tg $CH_4$ $yr^{-1}$ from inland waters (Saunois et al. 2019). Although these emission numbers have high uncertainties, aquatic systems are known to act as net sources (Bastviken et al. 2011; Raymond et al. 2013). Due to their significant $CH_4$ source strength, inland waters have seen an increase in attention (see Abril and Borges 2005; Panneer Selvam et al. 2014; Richey et al. 2002; Wang et al. 2009; Metlton et al., 2013; Zhang et al. 2018).

Wetlands are one of the single largest source within the inland waters (125–218 Tg $CH_4$ $yr^{-1}$) accounting for roughly one
third of total emissions (Dean et al. 2018; Saunois et al. 2019). They are usual intertwined with rivers, channels and lakes making them highly diverse regions. Rivers emit 1.5–26.8 Tg $CH_4$ $yr^{-1}$, and when combined with lakes contribute 73.1 Tg $CH_4$ $yr^{-1}$ (Bastviken et al. 2011; Stanley et al. 2016), although these numbers have large uncertainties (Kirschke et al. 2013).

Due to lakes being some of the easier systems to measure and compare, they are the most extensively covered components of inland waters and although only covering 0.9% of the Earth's surface, give a range of around 8–73 Tg $CH_4$ $yr^{-1}$ (Kirschke et
al. 2013). Specifically, shallow lakes are known to generally be hot spots in terms of $CH_4$ emissions (Cole et al. 2007; Davidson et al. 2018). Channels and rivers emit around 26.8 Tg $CH_4$ $yr^{-1}$ excluding ebullition (Stanley et al. 2016), however, due to a lack of global data coverage and consistency their role in both carbon transport and storage is not well constrained (Tranvik et al. 2009). Therefore, there is a need for more detailed assessment of the role of methane emissions from rivers and channels as they have been suggested to be more spatiotemporally variable for $CH_4$ than $CO_2$ (Stanley et al. 2016; Natchimuthu et al.
45 2017).

Typically, $CH_4$ is biogenically produced within anaerobic environments (see Fig. 1 for details), where microbial fermentation of organic matter occurs and is controlled by the interplay between input of organic matter and temperature (Stanley et al. 2016). This is generally the end of the line respiration process, through either hydrogenotrophic methanogenesis where oxidation of $H_2$ using $CO_2$ as a terminal electron acceptor produces $CH_4$ or by acetoclastic methanogenesis via the breakdown of simple
substrates or acetate, which is a major pathway within the fresh water systems (Whiticar et al. 1986). Other processes also include bubbles transport via ebullition accounting up to 50% of the total flux in certain systems, and generally contributing far larger than that of diffusive fluxes (van Bergen et al. 2019), or more physical processes such as vertical mixing, lateral transport and ground water inputs (Crawford et al. 2014a; Stanley et al. 2016).



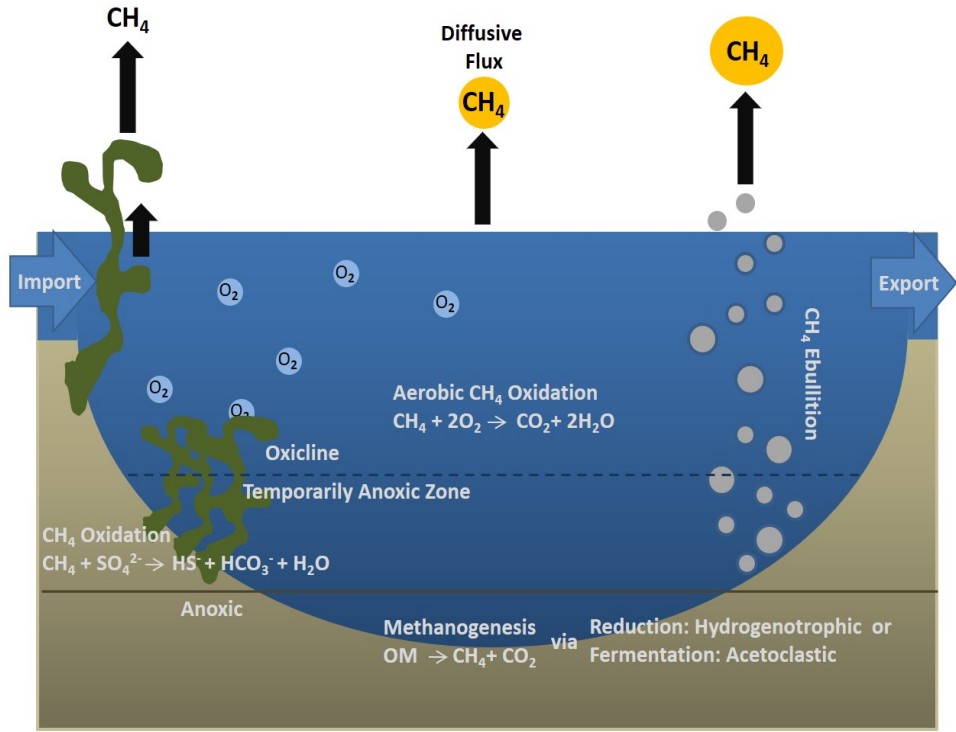

**Figure 1.** Some of the main processes within a lake in a typical stratified situation. Well-oxygenated surface waters are places of $CH_4$ oxidation. Macrophytes may release methane to the water column or directly into the atmosphere. The water column can host both methanogenic and methane oxidation pathways. Arrows pointing upwards symbolise the water-air fluxes resulting from typically observed supersaturation of methane in surface waters.

Diel cycles of dissolved gases within inland waters are driven by multiple processes including temporal variability of biolog-

ical processes such as photosynthesis and respiration, transportation, vertical stratification or temperature dependent solubility (Nimick et al. 2011; Maher et al. 2015; Zhang et al. 2018; Sieczko et al. 2020). Although potentially substantial, these are rarely considered in studies of $CH_4$ fluxes due to general lack of data. Just as with overall data coverage of $CH_4$, both spatially and temporally, there is also need for refined understanding of the contributions and the controls of $CH_4$ production and sources (Bogard et al. 2014; Abril and Borges 2019).

With climate warming, $CH_4$ production is set to increase from lakes as well as through eutrophication (Marotta et al. 2014; Del Sontro et al. 2019, Sepulveda-Jauregui et al. 2018). Bartosiewicz et al. (2019) suggest that increased warming and browning of the lakes will increase the warming of surface waters causing increasing stratification. This may lead to an increase the $CH_4$ production in bottom waters potentially leading to +8% of the current global lake emissions from shallow (< 5 m) lakes. Therefore, analyzing the spatial and temporal (i.e. at least seasonal and diel) variability of methane emissions is important

for future predictions and modelling efforts. Given the complexity of inland water systems, especially wetlands, monitoring approaches tend to stay within one system. Here we deployed an on-site monitoring device throughout the Danube delta,





which measured gas concentrations continuously from a moving platform. The acquired high spatial and temporal resolution of methane concentrations and corresponding emissions formed the observational data basis to assess the importance of different systems (lakes, rivers and channels) and of diel cyles for the overall methane emissions in such a complex system.

The Danube River Delta, as most river deltas, is known to be an important natural source of $CH_4$ (Cuna et al. 2008; Durisch-Kaiser et al. 2008; Pavel et al. 2009). Recently, Maier et al. (2020) investigated the seasonal emission rates of $CO_2$ and $CH_4$ in parts of the Danube Delta, focusing on a set of stations that were analyzed at monthly intervals. Here, we take a complementary approach focusing on extremely high-resolution data in space and at the diel time-scale, with focus on three field studies.

The objectives of this study are split into two main aspects: 1) to assess the differences between regions within the Danube

delta in regards to $CH_4$ concentrations and fluxes, and 2) to use high-resolution data to assess the importance of diel cycles both on a local and global scale in such systems.

## 2    Methods

### 2.1    Set up

A portable and versatile flow-through sensor set-up was placed on-board a small houseboat for continuous mapping through-

out the Danube Delta. Campaigns took place over three seasons: May (17–26), Aug (3–12), and Oct (13–23) 2017. The set-up consisted of the HydroC® $CO_2$ FT ($CO_2$ partial pressure, $pCO_2$, -4H-JENA engineering GmbH, Jena), HydroC® $CH_4$ FT ($CH_4$ partial pressure, $pCH_4$, -4H-JENA), HydroFlash® $O_2$ (dissolved oxygen, $O_2$, -4H-JENA) and a SBE 45 thermosalino-graph (Sea-Bird Electronics, Bellevue, USA) to measure temperature and conductivity. All sensors ran simultaneously at a speed of up to 1 Hz on the same continuous water flow (submersible pump deployed over the side at a depth of approx. 40

cm). The HydroC® $CO_2$ FT and the HydroC® $CH_4$ FT use non-dispersive infrared (NDIR) and tunable diode laser absorption spectroscopy (TDLAS) technology respectively, while the HydroFlash® $O_2$ Optode sensor uses the principle of dynamic flo-rescence quenching (Bittig et al. 2018). Further details on the setup, its calibration and validation can be found in Canning et al. (2020).

### 2.2    Study site

The Danube River Delta is located on the Black Sea coast of Ukraine and Romania (44°25'–45°30'N and 28°45'–29°46'E). Originating in Germany, the Danube River travels across 2,857 km, with a drainage basin of 817,000 $km^2$ (Panin 2003). The delta is a complex system of wetlands, lakes, rivers and channels, both manmade and natural, with the largest compact reedbed zone in the world (Oosterberg et al. 2000; Panin 2003). The fluvio-marine delta system accounts for 51% of the total area (Pavel et al. 2009) in which it sees salt intrusions and through-flow from the Black Sea into the delta. Since the 1970s, the Danube

Delta has been subject to eutrophication, with its peak during 1987–1988 (Cristofor et al. 1993; Galatchi and Tudor 2006; Enache et al. 2019). After a decrease of nutrient loads in the 1990's, due to socioeconomic changes in Eastern Europe, a slow





decline of nutrient levels was observed (Rîşnoveanu et al. 2004; Pavel et al. 2009), however, more recent levels comparable to those in 1988 were reported (Tudor et al. 2016; Spiridon et al. 2018).

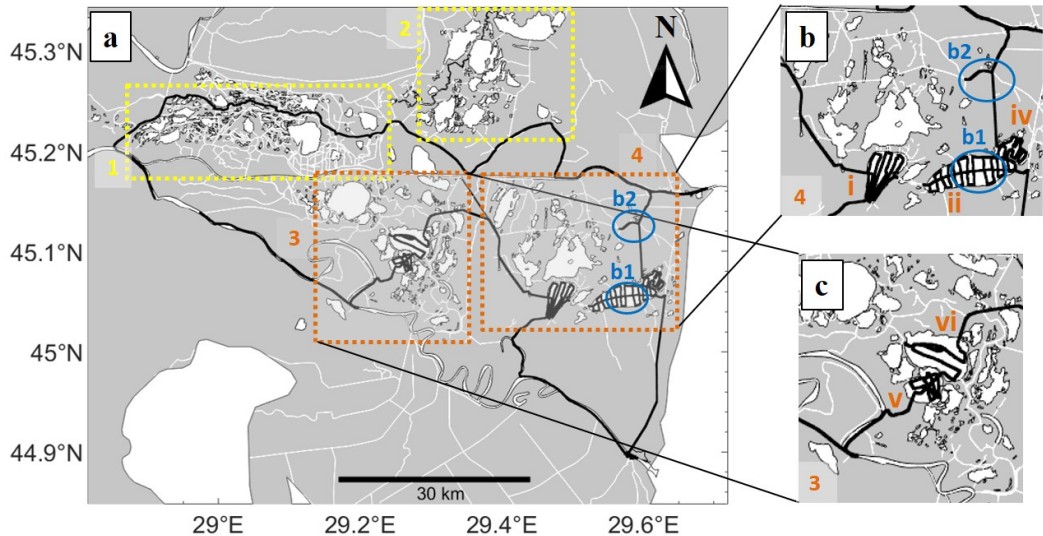

**Figure 2.** The Danube River Delta, Romania with the cruise tracks from the three seasonal campaigns in black, with only slight variations due to accessibility by boat. Systems of lake complexes in yellow: Contea-Furtuna (1), Matita-Merhei (2), Gorgova-Uzlina (3) and Roşu-Puiu (4), with only 3 and 4 extensively mapped and analysed here. Side figures show Lakes Puiu (b.i), Roşu (b.ii), Roşulette (b.iv), Uzlina (c.v) and Isac (c.vi). Blue circles indicate the sites of the two diel cycles at Lake Roşu (b1) and the 'hot spot' channel (b2), both during the August campaign.

The delta is within the temperate climate system, but experiences extreme temperature ranges with air temperature from below freezing (0°C) to +30°C (ICDP 2004). Deltas are continuously changing landscapes, with moving lake systems and floating islands. The overall Danube delta is roughly 4423 km$^2$ with a 67–81% coverage in either aquatic ecosystems (rivers, lakes and channels) or wetlands (Cristofor et al. 1993). Using the small houseboat, the set-up was fixed, and a thorough transect throughout the delta was carried out with extensive lake transects completed in all three seasons for comparability (Fig 2). This study also featured two stationary diel cycle measurements (Fig 2b: blue circles), one in Lake Roşu and the other in the channel where we witnessed a major biogeochemical 'hot spot'.

## 2.3 Computations of saturation and fluxes

The average global atmospheric CH$_4$ concentration (ppb) was taken from NOAA/ESRL Global Monitoring Division program (Dlugokencky 2020) for May, Aug and Oct 2017 (1847, 1844.7 and 1858.1 ppb respectively). As the delta is practically sea level, barometric pressure as well as wind speed measured at the Gorgova station were used. Schmidt numbers (Sc) were calculated for temperature dependence following (Wanninkhof 1992) for freshwater. The corrected Schmidt numbers varied between 296 and 824 in this study, consistent with the large temperature variance. Using CH$_4$ solubility (Wiesenburg and





Guinasso 1979), $CH_4$ equilibrium concentrations in water were calculated and employed in the flux calculation. Fluxes were calculated following Peeters et al. (2019; supplementary material S3.2). Given slow stream velocities, we used the parameterisation from Cole and Caraco (1998) with constant gas-transfer velocity of $\sim$2 cm h$^{-1}$ in the absence of wind

$$\text{k}_{600} = 2.07 + 0.215 \cdot \text{U}^{1.7} \quad \text{cm} \quad \text{h}^{-1} \tag{1}$$

where $U$ is wind speed at 10 m height in m s$^{-1}$, and $k_{600}$ is the gas transfer velocity normalised to a Schmidt number of 600, i.e. $CO_2$ in freshwater at 20°C (Jähne et al. 1987; Crusius and Wanninkhof (2003):

$$\text{k}_{CH_4} = \text{k}_{600} \left(\frac{\text{Sc}_{CH_4}}{600}\right)^{\text{n}} \tag{2}$$

$$for\ U \leq 3.7\ m\ s^{-1}\ n = -\tfrac{2}{3},\ for\ U > 3.7\ m\ s^{-1}\ n = -\tfrac{1}{2}$$

where $k_{CH_4}$ is the transfer velocity at $Sc_{CH_4}$, which is the Schmidt number of $CH_4$ at a given temperature, and the exponential $n$ reflects two wind speed regimes (Jähne et al. 1987). For rivers, due to differing fetch and dynamics we used $n =$ -0.5 throughout, consistent with multiple river studies (Borges et al. 2004; Guérin et al. 2007; Bange et al. 2019). The flux was then calculated using the $CH_4$ concentration in the water and air:

$$\text{Flux} = \text{k}_{CH_4} \cdot (\text{CH}_{4,\text{water}} - \text{CH}_{4,\text{air}}) \quad \text{mol} \quad \text{m}^{-2} \quad \text{s}^{-1} \tag{3}$$

Given that the effect of spatial variability of $k_{CH_4}$ is relatively small in lakes with surface areas of 5x10$^5$ m$^2$ or larger, we disregarded size effects of lakes on emission fluxes noted by Schilder et al. (2013). In the following analyses, both day and night data will be shown unless stated otherwise for $CH_4$.

## 3 Results and discussion

### 3.1 Concentrations distribution and estimated fluxes

#### 3.1.1 Overall $CH_4$ situation in the Danube Delta

Our high spatiotemporal resolution $CH_4$ data showed constant supersaturation ($CH_4$ concentration range 113 to 15600 nmol L$^{-1}$), throughout the delta. Both significant systemical and seasonal variability was observed, with channels having the highest concentrations of up to 15600 nmol L$^{-1}$ (Table 1) and showing overall a magnitude higher values compared to rivers. The concentrations are within the lower ranges previously observed (20 to 280000 nmol L$^{-1}$) for $CH_4$ in oxic freshwaters (Tang et al. 2016; Bižić-Ionescu et al. 2019).

High spatial variability was found across systems and water type boundaries (such as channels to lakes), which was also observed clearly by Crawford et al. (2017). More confined areas in closer proximity to the wetlands, were found to have the





highest concentrations. These boundary crossovers were due to seasonal changes in concentrations and change of flow direction

varying throughout the delta. May and Aug show similar median $CH_4$ concentrations at 627 nmol $L^{-1}$ and 951 nmol $L^{-1}$, respectively, Oct $CH_4$ median level however, had increased to 1440 nmol $L^{-1}$ (Fig. 3). In each season, 3 specific locations stood out with extreme $CH_4$ concentrations: the two channels joining Lake Puiu (Crisan channel to the north and one from the south), and the 'hot spot' channel anomaly (blue circle (b2) on channel in Fig. 2). Rivers and channels (including the anomaly) showed the highest variability during Aug and May, consistent with the directional flow regime bringing in the water from the

surrounding wetlands after the flood waters. The highest median was during Oct for rivers, lakes and channels (median: 559, 693 and 1500 nmol $L^{-1}$ respectively), which coincides with the degradation of the macrophytes.

Oxygen ($O_2$) was mostly undersaturated, however measurements were not distributed proportionally throughout the delta potentially leading to the lower median in May from more measurements collected in the 'hot spot'. During Aug and Oct, $O_2$ saturation (%) was generally above 60% with Aug showing the larges variability above 100% coinciding with both temperature

and production. These values included the 'hot spot', given it is a natural feature and most likely not the only one within the Danube delta.

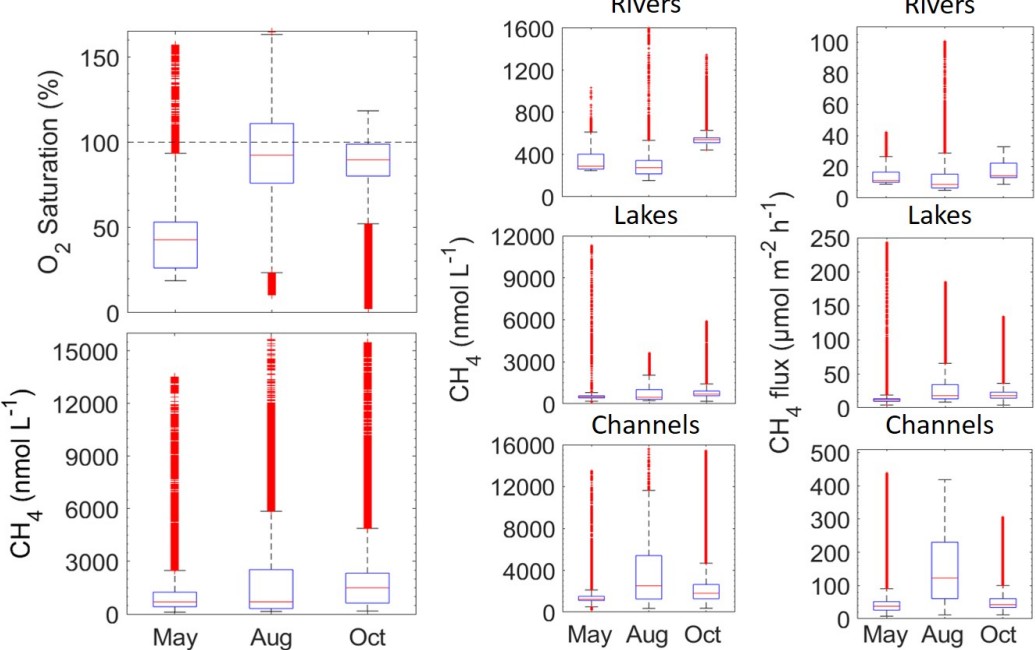

**Figure 3.** Left panels: oxygen saturation and $CH_4$ concentration data from the three seasonal campaigns: May, August, and October. Central panels: $CH_4$ concentrations split by water type (rivers, channels and lakes). Right panels – $CH_4$ fluxes by water type. The blue boxes represent lower (25%) and upper (75%) quartiles, with the whiskers marking the lowest and highest data point within a distance of 1.5 times the interquartile range from the respective quartile. Red '+' signifies points outside of these boundaries. Median values are shown as horizontal red lines. All data are included (n > 200,000 for each season), including the hot spot and day/night cycle data.





Concentrations almost translate to the water-air fluxes (Fig. 3), with some variability ultimately due to temperature and wind. Using the estimated area from Maier et al. (2020) for total area of rivers, channels and lakes (164, 33, 258 km$^2$ respectively) we estimated the delta's total emissions. Taking the average across all 3 seasons we got an overall mean outgassing flux density of $49 \pm 61$ $\mu$mol m$^{-2}$ h$^{-1}$. This gave an emission range of $0.00000198 - 0.00000538$ Tg yr$^{-1}$ CH$_4$ for the combined region covered by rivers, lakes and streams (455 km$^2$). We used the above mean calculated flux for the area and applied this to the total area over the entire year. The global overall average estimate for wetlands, lakes, streams and rivers of 117–212 Tg CH$_4$ yr$^{-1}$ (Saunois et al. 2019), from a total area of 596–894 million ha translates to an average global emission of $140 - 170$ $\mu$mol m$^{-2}$ h$^{-1}$ about a factor of three higher than the flux rate obtained here for the Danube Delta. However, this estimate assumes diffuse emissions only.

Diffusive fluxes have been suggested to underestimate total fluxes from lakes by 277% (Sanches et al. 2019) due to mechanisms like micro-bubble flux enhancement, ebullition and direct plant emissions (Carmichael et al. 2014; McGinnis et al. 2015). In their study over two years, however, Maier et. al., (2020) found evidence that bubble emission of methane in the Danube Delta lakes and channels, potentially accounted for $\sim 70$ %. CH$_4$ fluxes calculated in this study were within the ranges of diffusive flux measurements reported by Maier et al. (2020) for rivers and lakes, whereas channels were within the range observed using their total fluxes (diffusive and potential ebullition fluxes), exceeding that of purely diffusive flux measurements. Median lake measurements within this study were about 63 % lower than that reported by Maier et al. (2020), using total fluxes. This coincides well with the $\sim 70$ % accountablity for ebullition fluxes from lakes.

Although measured within the same regions, methods from both studies are not comparable due to the use of floating chambers at a few specific location, potentially missing highly variable spatial variation and explaining the far larger diffusive fluxes found here in channels (Maier et al (2020): 0.16–6.2 mmol m$^{-2}$ d$^{-1}$, this study: 0.18–10.5 mmol m$^{-2}$ d$^{-1}$ ).



**Table 1.** Statistics with minimum (Min), median (Med), maximum (Max) of $CH_4$ concentrations (nmol $L^{-1}$), $CH_4$ saturation (%) and $CH_4$ flux ($\mu$mol $m^{-2}$ $h^{-1}$) for rivers (R), channels (Ch) the hot spot channel (HS) and lakes (L) over the 3 seasons: May, Aug and Oct 2017.

| System | | $CH_4$ nmol $L^{-1}$ | | | $CH_4$ Saturation % | | | $CH_4$ Flux $\mu$mol $m^{-2}$ $h^{-1}$ | | |
|---|---|---|---|---|---|---|---|---|---|---|
| | | **May** | **Aug** | **Oct** | **May** | **Aug** | **Oct** | **May** | **Aug** | **Oct** |
| **R** | **Min** | 248 | 154 | 441 | 8560 | 6420 | 14200 | 8.9 | 4.9 | 8.8 |
| | **Med** | 302 | 290 | 541 | 10500 | 12200 | 17600 | 11.2 | 8.7 | 14.2 |
| | **Max** | 1040 | 1610 | 1350 | 35600 | 66600 | 44600 | 42.1 | 101 | 33.9 |
| **Ch** | **Min** | 221 | 355 | 369 | 7990 | 14000 | 11800 | 7.6 | 11.6 | 11.9 |
| ***** | **Med** | 1170 | 1300 | 2230 | 40900 | 55700 | 71400 | 39.1 | 63.4 | 49.4 |
| | **Max** | 6950 | 4270 | 6120 | 242000 | 180000 | 203000 | 225 | 220 | 166 |
| **HS** | **Min** | 1030 | 1830 | 994 | 34000 | 78600 | 32700 | 22.8 | 82.4 | 19.7 |
| | **Med** | 1210 | 5710 | 1730 | 40400 | 237000 | 56900 | 32.2 | 211 | 36.1 |
| | **Max** | 13500 | 15600 | 15500 | 469000 | 631000 | 507000 | 438 | 418 | 306 |
| **L** | **Min** | 113 | 224 | 177 | 4120 | 9450 | 5590 | 3.9 | 8.3 | 3.9 |
| ****** | **Med** | 465 | 466 | 693 | 16100 | 19300 | 22300 | 11.8 | 17.8 | 17.8 |
| | **Max** | 11300 | 3650 | 5930 | 395000 | 166000 | 187000 | 244 | 185 | 135 |

\* Excluding Stinky channel ('hot spot') and connecting channels, due to this location experiencing extremely high concentrations as an 'anomaly' within our full transect

\*\* Higher range in lakes due to influence of channels

From our meteorological data, we found little correlation with external factors such as wind, however, given these were not measured in situ, this cannot be fully quantified. We therefore suggest the observed distribution patterns over the entire delta

are mostly more driven by both biological and physical processes affecting water-side $CH_4$ concentrations instead of effected by external factors, as previously suggested (Bange et al. 2019; Sanches et al. 2019) where precipitation was potentially responsible for a decease in concentrations.

### 3.1.2 Seasonality

Seasonally, the delta changes significantly owing to a range of processes. High concentrations and therefore fluxes during

May, have previously been explained due to growth, temperature and biomass peak, linking plant biomass to $CH_4$ emissions during growing season (Milberg et al. 2017). This can be further linked to the previous flood period just before the transect in May. The Danube delta is known to have high levels of nutrients (Panin 2003; Durisch-Kaiser et al. 2008; Spiridon et al.





2018) arriving from the Danube river. This could account for higher concentrations, and saturation due to enhanced plankton growth being a source of additional labile organic matter fuelling $CH_4$ productivity in the sediments, which then outfluxes

(Mendonça et al. 2012; Ward et al. 2017). Aug showed the largest $CH_4$ range among the seasons, however with the lowest median coinciding with the theory that there is decreased $CH_4$ concentrations and emissions during lower water levels (Melack et al. 2004; Marín-Muñiz et al. 2015; McGinnis et al. 2016).

During Aug and Oct, the process of macrophyte degradation within the delta, mainly the lakes, was linked with elevated $CH_4$ concentrations in specific locations. This sharp increase of biodegradable organic matter triggered anoxic decomposition

of organic carbon which released $CH_4$ (Segers 1998).

The channels are highly influenced by the surrounding reed beds, which are known to produce high levels of $CH_4$ (Bastviken et al. 2011), and have influence on the surrounding systems they flow into (e.g. lakes). This both explains the high variability (Fig. 3) and higher overall concentrations and fluxes (Table 1). They are also influenced by the lakes, which are sources of labile organic carbon that fuels methanogenesis. However, given methanogenesis was not measured, we can only make assumptions

about this. Channels are the links between the rivers and the lakes, surrounded by wetlands form which the collect water and therefore generally have the highest concentrations of $CH_4$ and lowest in $O_2$. Delta systems are highly diverse, and therefore each region has been split to give a more descriptive assessment of the dynamics in the Danube delta.

### 3.1.3 'Hot Spot'

The 'hot spot' was classified as a small channel system receiving partially anoxic water from the reed stands (Fig. 2b (b1)).

The highest conductivity was observed around the 'hot spot' as 0.08 S m$^{-1}$ (overall mean $\pm$ SD of 0.038 $\pm$ 0.005 S m$^{-1}$), suggesting also the potential of ground water influences. Given the dramatic change within the concentrations and properties of the water, such as the water temperature dropping further into the 'hot spot', even within summer, this would further provide evidence from cooler ground waters or potential waters from the reed beds also suggested by Maier et al. (2020).

The 'hot spot' showed seasonality in concentrations and dynamics. In Aug, median fluxes measured 211 $\pm$ 86.3 $\mu$mol m$^{-2}$

h$^{-1}$, however when compared to all months combined, the median from the 'hot sport' reduced to 54.9 $\pm$ 106 $\mu$mol m$^{-2}$ h$^{-1}$. The influence of the 'hot spot' on the surrounding areas was significant, with high concentrations tending to diperse into the following channels (Canning et al. 2020). However, the influence of the 'hot spot' on the data as a whole system, is more dependant on the extension of this location. In the recent study by Maier et al. (2020), it was estimated that due to other similar environments within the delta, areas of little movement, could account for 2% of the total channel area, or 20 % of $CO_2$ and

$CH_4$ fluxes from the channels.

### 3.1.4 Fluvial $CH_4$

The fluvial delta (rivers and channels) works as the supply of incoming water into the main part of the delta, accounting for the base level of $CH_4$ concentrations being laterally transported. We found very little evidence that intrusions from the Black Sea may have reached into the delta and have an impact such as suggested before (Durisch-Kaiser et al. 2008; Pavel et al.

2009). This would be important to explain reduced methane production as sulfate reduction becomes the dominating anaerobic





mineralization. Rivers had the lowest range of concentrations for $CH_4$ with the smallest variability out of all systems and the delta (Fig. 3) ranging from 154 to 1600 nmol $L^{-1}$ (during Aug). In channels, when excluding the 'hot spot', medians were larger than rivers but fairly consistent throughout May and Aug yet higher during Oct (1170, 1300 and 2230 nmol $L^{-1}$ respectively) within channels. However, channels experienced some of the highest concentrations when including the 'hot spot', ranging

from 221 to 15600 nmol $L^{-1}$ for May and Aug respectively. It also changed the dynamics during Aug, observing the highest median of 5710 nmol $L^{-1}$, showing the significant influence one hot spot can have on a system. This provides evidence that most of the $CH_4$ production happens within the delta, not the river itself.

As stated before, $CH_4$ fluxes followed roughly the same trend as $CH_4$ concentration, only moderately modulated by variable wind speed. For rivers, such as with concentrations, Aug fluxes had the highest variability (Table 1 and Fig. 3) spanning from

4.9 to 100.7 $\mu$mol m$^{-2}$ h$^{-1}$ $CH_4$, however had the lowest median of the seasons (8.7 $\mu$mol m$^{-2}$ h$^{-1}$). Comparing Oct to May and Aug for rivers, it had the largest percentile range and median (14.2 $\mu$mol m$^{-2}$ h$^{-1}$). Channel fluxes from all months combined had a median of 47.9 $\pm$ 70.6 $\mu$mol m$^{-2}$ h$^{-1}$, higher than both May and Aug alone (39.1 and 49.4 $\mu$mol m$^{-2}$ d$^{-1}$: excluding the 'hot spot'). This is potentially linked to the increased degradation of macrophytes and other organic matter during Oct as stated before.

Overall our calculated mean flux for all months combined from the fluvial delta was 594 $\pm$ 525 $\mu$mol m$^{-2}$ h$^{-1}$, within the diffusive mean from the overall literature (342.5 $\pm$ 1062.5 $\mu$mol m$^{-2}$ h$^{-1}$; Sanley et al. 2016). However, we found a far higher median of 473 $\mu$mol m$^{-2}$ h$^{-1}$ (compared to 33.3 $\mu$mol m$^{-2}$ h$^{-1}$). The fluvial delta had a mean of 2030 $\pm$ 2110 nmol $L^{-1}$, with a median (1520 nmol $L^{-1}$) comparable to that of Stanley et al. (2016) with a mean of 1350 $\pm$ 5160 nmol $L^{-1}$. When comparing within the fluvial system (rivers and channels separately), riverine $CH_4$ concentration during May and Aug had a

median comparable to channels and therefore showed overall homogeneity, however channels appeared to have more extreme values and ranges than rivers. This difference would be due to less biological and physical processes occurring within the rivers due to depth, proximity to the wetlands and the flow generally being faster. However, both rivers and channels concentrations varied, showing large dependence on both seasonal changes and sample location. Furthering evidence, just as with the 'hot spot', for significant spatiotemporal influence on $CH_4$ fluxes.

### 3.1.5 Situation of $CH_4$ in lakes

Lakes showed concentrations similar to those of Pavel et al. (2009) (see appendix A1), although taken roughly 10 years later. The comparison to this earlier study indicates, that eutrophication and carbon turnover have not significantly changed during this period (Tudor et al. 2016; Spiridon et al. 2018). These concentrations ranged from the lowest 113 nmol $L^{-1}$ to the highest 11300 nmol $L^{-1}$ both in May (largest concentration close to a channel). The median however, stayed roughly the same for

both May and Aug (465 and 466 nmol $L^{-1}$ respectively), with Oct reaching 630 nmol $L^{-1}$. We expect less productivity and more mineralization of macrophytes in Oct, leading to enhanced $CH_4$ production. Before entering each lake complex, the water had to travel through either the channels or the reed beds, increasing the concentrations coming into the lakes. The inflowing water however quickly dispersed, and was soon oxidized as seen before (Crawford et al. 2017). This inflow was only visible on the edges of the lakes and although had influence on the overall concentration, were seen as outliers as the $CH_4$ due to





being quickly oxidised (Fig, 4). Morphology and seasonal changes were far clearer in the lakes than any of the other systems, due to noticeable influences from the channels showing larger productivity and macrophyte distributions. This led to higher concentrations during Oct as the macrophytes broke down as mentioned before, but also potential stratification (Milberg et al. 2017).

By averaging over the lakes, we obtained the total lake area fluxes of 2.9, 6.5 and 4.8 mol $CH_4$ $h^{-1}$ for May, Aug and Oct
respectively. Sediment fluxes are one of the main sources of $CH_4$ diffusion fluxes (Peeters et al. 2019), however, among other paths, ebullition can also be a significant source of $CH_4$ and its impact on the fluxes can vary from both system and location (see Bastviken et al. 2008, McGinnis et al. 2016, Schubert and Wehrli 2019 and van Bergen et al. 2019 for varying quantities). As it is not possible to capture ebullition through dissolved $CH_4$ surface measurements, such as in this study, this can potentially lead to mild-significant underestimations (Maier et al. 2020). However, the benefits of this study, were being able to pick up
local dynamics that is usually missed by just daily or intermediate sampling.

### 3.2   Diel $CH_4$ cycling

One advantage to measuring continuously at high-resolution, was the opportunity to observe diel cycles. These extractions of temporal variability (i.e. over nearly a full 24 h cycle (Fig. 4)) were successfully carried out at specific locations. For analyses and comparison, two diel cycles were recorded: one in Lake Roşu (Fig. 2b(ii)), and the other within the 'hot spot', both
locations <3 m depth.

Lake Roşu's diel cycle (Fig. 4 left) shows clear indications of strong temporal variability on the diel time scale. The nocturnal buildup in $CH_4$ is linearly correlated with the loss of oxygen (molar $CH_4$:$O_2$ ratio 1:-50). $CH_4$ concentrations started from 400 nmol $L^{-1}$ at sunset and reached 1400 nmol $L^{-1}$ at sunrise. During the diurnal period, $CH_4$ concentrations quickly relaxed back to initial conditions. As the mapping transect in Lake Roşu started already around 9:00, some spatial variability is superimposed
from then on to the dominant diel cycle causing $CH_4$ concentrations to vary over the range 200–500 nmol $L^{-1}$. Overall, the $CH_4$ concentration shows a strong co-variation with oxygen. The diurnal relaxation of the $CH_4$ and $O_2$ concentrations to initial state has a more exponential shape. A possible explanation for this hysteresis: the water column stratifies during the day, and undergoes vertical mixing as the surface water is cooling during the night. This process progressively mixes the two formerly separated water bodies resulting in the observed linear mixing line (Milberg et al. 2017). Diurnal warming then
quickly re-stratifies the water column so that the surface layer has no further entrainment from low-oxygen, high-methane waters below and undergoes rapid $CH_4$ loss due gas exchange (Fig. 4). In contrast to oxygen, $CH_4$ does not reach equilibrium during the diurnal period. This could be due to continued supply from background sources of $CH_4$ (e.g. from macrophytes, lateral transport, diffusive flux across the thermocline or production via photoautotrophs (Bižić et al. 2020). Given the rate and extent of the $CH_4$ increase, this shows potentially significantly $CH_4$ production during the day in the bottom waters (Grasset
et al. 2019), supporting the hypothesis of anoxic conditions close to the sediment and therefore intensified methanogenesis (Crawford et al. 2014b; 2017). This is more likely to lead to other effective transport of $CH_4$ such as ebullition which could supply $CH_4$ to the surface waters or the atmosphere. Oxygen, in contrast relaxes back to equilibrium during the day as both air-water fluxes and in-situ photosynthetic production of $O_2$ would drive the system towards equilibrium.



**Figure 4.** CH$_4$ (a + b), CO$_2$ (c + d) and temperature (e + f) against O$_2$ concentration as measured during diel cycle experiments in lake Roşu (left column) the 'hot spot' (right column). Colour bar denotes time of the day (hh:mm). Sunrise and sunset are also indicated. Both studies were carried out during the Aug (summer) campaign. During the night from just before 20:00 until 09:00, the boat was anchored and stationary. Transects through the following day continued to map the lake, whereas the channel was all in one anchored location.



The data (Fig. 4c) also show a clear hysteresis in the relationship between $CO_2$ and $O_2$ changes over a diel cycle. The $CO_2$

peak of about 8 $\mu$mol L$^{-1}$ (corresponding to a p$CO_2$ of about 250 $\mu$atm) just after sunrise is around 65% saturated, only

during the transect do some values exceed 100%, reaching 130% (Fig. 4c few points over 15 $\mu$mol L$^{-1}$). The data show a

slight decoupling of metabolism of $CO_2$ and $O_2$ (Fig. 4d), such as $CO_2$ increasing significantly without the use of $O_2$, which

has also been observed by Peeters et al. (2016). These concentrations, however, coincide with the mapping; higher $CO_2$ rates

when closer to the lake edges (similar to the $CH_4$ pattern in Fig. 5), due to incoming waters from the wetlands. During the

day, Lake Roşu is undersaturated in $CO_2$ and supersaturated in $O_2$, indicating high levels of productivity in the surface waters.

Overnight we observe respiration with the $CO_2$ increasing towards equilibrium and $O_2$ moving away from equilibrium. This is

an indication for high rates of primary production during the day with an intense drawdown of $CO_2$ which is not compensated

during the night, as has been observed in eutrophic lakes (Balmer and Downing 2011). Our observed $CO_2$ concentrations were

far lower than those reported by Pavel et al. (2009): 26 $\pm$ 27 $\mu$mol L$^{-1}$ during September 2006.

The 'hot spot' (Fig. 4, right) also shows a clear co-variation of $CH_4$ with oxygen. Here $CH_4$ increases from roughly 4000

to 16000 nmol L$^{-1}$ over the nocturnal period (sunset to sunrise), followed by a rapid return to values around 6000 nmol L$^{-1}$

during the diurnal period (sunrise to sunset). $O_2$ decreases while $CH_4$ stays roughly the same until around 3:30 am when it

appears to enter into hypoxic and even towards suboxic conditions as the ratio increases to about 1:3. This pronounced non-

linearity be indicative of mixing with more than two endmembers, e.g., surface layer, sub-surface layer and a distinct bottom

layer. The initial mixing encompasses only surface and sub-surface layer (similar to the lake situation) whereas later during the

night, near-bottom waters are entrained that have extremely elevated $CH_4$ concentration (and no oxygen) as a consequence of

anoxic methanogenesis in sediment pore waters. An alternative explanation would be groundwater or lateral injection of water

from adjacent wetlands.

$CO_2$ reaches saturation levels of close to 4500% during the diel cycle in the 'hot spot', over the night with the lowest

supersaturation of 1175% at the end of the diel cycle ($\sim$150 to 550 $\mu$mol L$^{-1}$). Dissolved $CO_2$ displays a mirror image with

temperature (Fig. 4f). The $CO_2$:$O_2$ relationship has a molar ratio 2:1 (with indications of progressive steepening as observed

more clearly for $CH_4$) during the night and such as with $CH_4$, $CO_2$ shows a steep decrease following sunrise, with initially

little change in $O_2$. The diel hysteresis is far clearer with $CO_2$ than with $CH_4$, showing a steady increase and decrease. This

is ultimately due to different processes, and potential methanogenesis occurring in the bottom waters before mixing, as stated

above.

The diel changes in temperature are roughly the same for the two situations ($\pm$ 2.5°C: Fig. 4), showing influence on all

variables and induced strong density variations. The observed strong density variations were potentially sourced by the mixing

of the bottom waters over the course of the night (Fig. 4), when cooling of the warm surface layer mixed with the colder

bottom waters. Although it could be argued that temperature could have had an effect within the diel variability as previously

suggested (Yvon-Durocher et al. 2014), temperature variability only causes a 3% change in methane solubility. Compared with

the variability over the night, the transect during the day that covered the entire lake showed $CH_4$ generally staying consistent

once the sun rose ($\sim$200–400 nmol L$^{-1}$ with peaks due to shorelines), which is roughly the same concentration as the previous

day, such as with all other variables. Statistically we also found no correlation between temperature and $CH_4$ flux (van Bergen





et al. 2019) over the entire lake ($P > 0.05$), therefore showing our diffusive fluxes are more reliant on the internal processes of

the water.

**Table 2.** Mean concentrations ($\pm$ SD) as well as hourly and yearly fluxes within Lake Roşu and the 'hot spot' channel with the exclusion of the night data (day light only: DL, Fig. 5 map) and all data (full diel cycle: FD, Fig. 5 all black and red data) for $CH_4$ flux, $CH_4$, $CO_2$ and $O_2$ concentrations.

| Mean $\pm$ SD | | | Lake Roşu | | 'Hot spot' channel | |
|---|---|---|---|---|---|---|
| **DL CH$_4$** **Flux density** | $\mu$**mol** **m$^{-2}$ h$^{-1}$** | **mol** **m$^{-2}$ y$^{-1}$** | $\mathbf{18 \pm 6.2}$ | $\mathbf{0.16 \pm 0.05}$ | $\mathbf{263 \pm 65.4}$ | $\mathbf{2.3 \pm 0.6}$ |
| FD CH$_4$ Flux density | $\mu$mol m$^{-2}$ h$^{-1}$ | mol m$^{-2}$ y$^{-1}$ | $19 \pm 9.7$ | $0.17 \pm 0.08$ | $224 \pm 85$ | $2 \pm 0.7$ |
| **DL CH$_4$ nmol L$^{-1}$ concentration** | | | $\mathbf{471 \pm 148}$ | | $\mathbf{7600 \pm 2630}$ | |
| FD CH$_4$ nmol L$^{-1}$ concentration | | | $530 \pm 300$ | | $6820 \pm 2950$ | |
| **DL CO$_2$ $\mu$mol L$^{-1}$ concentration** | | | $\mathbf{4.3 \pm 1.8}$ | | $\mathbf{307 \pm 125}$ | |
| FD CO$_2$ $\mu$mol L$^{-1}$ concentration | | | $4.8 \pm 1.8$ | | $315 \pm 120$ | |
| **DL O$_2$ $\mu$mol L$^{-1}$ concentration** | | | $\mathbf{217 \pm 22.7}$ | | $\mathbf{78 \pm 47.1}$ | |
| FD O$_2$ $\mu$mol L$^{-1}$ concentration | | | $265 \pm 23.5$ | | $105 \pm 60.3$ | |

To show the impact of these diel cycles, Table 2 summarizes the mean $CH_4$ concentrations and fluxes from the transect ($\sim$ 09:00 until 17:20, Fig. 5) and from the entire diel cycle (almost 24 hours: $\sim$ 18:55 8th Aug 2017 until 17:20 9th Aug 2017). The mapping route is representative of a high spatial resolution mapping routine (Fig. 5). The diel cycle was observed within

the mapping transect and therefore we were able to extract this section (Fig. 5c). Fluxes from the transect during the day (DL) and the full diel cycle (FD) were then scaled up to year averages showing an underestimation by just day night data alone. For the 'hot spot',we used the day night data (after sunrise) for this comparison due to no mapping transect following the diel cycle.

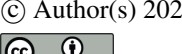



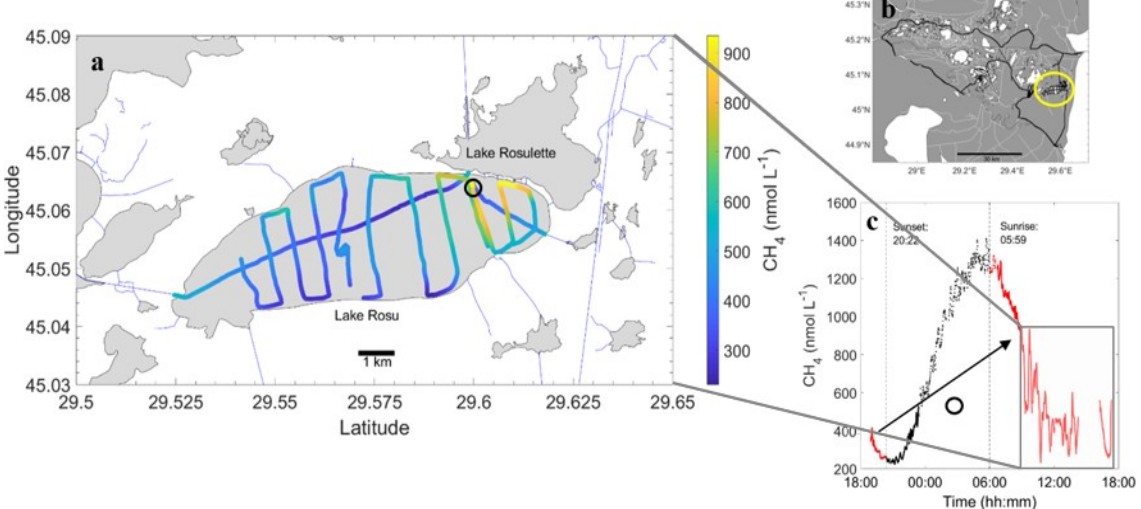

**Figure 5.** Mapping transect of lake Roşu (a), with the stationary location (c: circle), map the entire Danube delta with lake Roşu (b: yellow circle), and $CH_4$ concentration over time to show the distribution over the entire diel cycle (b: black showing between sunset and sunrise, red is day-time data, and box showing mapping transect). These were used to calculate the day light only data (b) and combined data (b: full data), with the diel cycle from lake Roşu (b: box) used for the day night data. Location of diel cycle during the night shown in a: black circle
.

Overall, the values for day-light (DL) period and full day (FD) show little difference due to the fact that the daily extremes

are encountered around sunrise and sunset such that full coverage of the daylight hours captures the full dynamic range of the diel cycle. An observation strategy based on continuous underway measurements could therefore minimize biases from any diel cycle variability even if measurments are restricted to the daylight period. However, even extensive mapping throughout the day is not representative of typical sampling methods which are mostly based on discrete samples taken at some time during daylight hours. A spot sampling approach during daylight hours could, depending on the timing, thus fall somewhere in the

entire diel peak-to-peak amplitude (approx. 300 - 1300 nmol $L^{-1}$ in Lake Roşu and 2500 - 14500 nmol $L^{-1}$ in the hot spot channel). Spot sampling without any knowledge of diel cycle variability therefore incurs an uncertainty range of a factor of 4.5 in this particular case. This result clearly calls for a sampling strategy that takes diel variability into account. Spot sampling near sunrise, sunset and during mid-day would appear to be a minimum requirement for representative results.

Excluding all full diel cycles from the entire data set, the mean $CH_4$ flux decreased from $49 \pm 61$ $\mu$mol m$^{-2}$ h$^{-1}$ to 34.9

$\pm 35.7$ $\mu$mol m$^{-2}$ h$^{-1}$, or a factor of 1.4. Therefore, scaling this by year changes the fluxes for the entire Danube delta from $0.4 \pm 0.5$ mol m$^{-2}$ h$^{-1}$ to $0.3 \pm 0.3$ mol m$^{-2}$ h$^{-1}$. Aug showed the largest variability when extracting diel cycles, with an uncertainty range of a factor of 2.27 from $84 \pm 38$ to $37 \pm 33$ $\mu$mol m$^{-2}$ h$^{-1}$. This greater variability can be linked to higher temperatures, greater stratification and increased production and organic matter degradation, all leading to potential increases in $CH_4$ (Duc et al. 2010; Fuchs et al. 2016). However, given diel cycles were not continuously measured throughout the entire

system, these values may not express the exact change when including all data from both day light hours and at night. To note,





upscaling these numbers is to give more of an awareness of the difference between including and excluding diel cycle data and between spot sampling. These values should not be taken at face value due to the major component of the Danube delta in terms of $CH_4$ influence (the wetland) was not measured and only diffusive fluxes calculated.

There have been multiple studies looking into diel cycles (Nimick et al. 2011; Zhang et al. 2018; van Bergen et al. 2019;
Sieczko et al. 2020), yet they are usually undetected or unresolved therefore are often ignored, particularly in studies with sampling during daylight hours. This can lead to substantial under- or overestimation of emissions, as has also been noticed in systems with high $CH_4$ concentrations (Natchimuthu et al. 2017). From our evidence of lake diel cycles, we conclude, in terms of discrete sampling techniques to fully capture the full variable cycle, sampling should be conducted during 3 periods: before sunrise, just after sunset and during the early afternoon. This way there is potential to capture the peak amplitude, low and
average concentrations, providing a better overall estimate of concentrations. Although, the best sampling techniques would be mapping with complete spatiotemporal coverage, however this is unfeasible in most cases.

Typically, delta systems tend to be either measured in specific regions (entrances or middle of lakes or channels), or with in situ measurements over time (e.g. Cuna et al. 2008; Wang et al. 2009; Olsson et al. 2015; Cunada et al. 2018). These measurements are then usually upscaled from single locations (e.g. Bouillon and Dehairs 2007; Borges et al. 2015; Joesoef et
al. 2017), failing to include spatial variability, system specific impacts (such as the 'hot spot' we observed here), and monthly changes. Here we can see that all of these impacts can have significant effects on the observed measurements. In Table 2 and Fig. 5, sampling time clearly has an impact, especially for upscaling.

## 4 Conclusions

To conclude, the overall Danube river delta surface waters were a source of $CH_4$, at a mean concentration of $1700 \pm 1930$ nmol
$L^{-1}$ and calculated aquatic emission to the atmosphere of $0.43 \pm 0.53$ mol m$^{-2}$ yr$^{-1}$. This is comparable to concentrations and diffusive flux mean of other systems of this type and size (see Stanley et al. (2016) for literature comparison: $1350 \pm 5160$ nmol $L^{-1}$ and $3 \pm 9.3$ mol m$^{-2}$ yr$^{-1}$ and Maier et al. (2020)). However, given that wetland systems (and therefore the reed beds) are known to be the significant in $CH_4$ fluxes of high variability (Segers 1998; Nisbet et al. 2019), our data only cover the water-air interface of channels, rivers and lakes and therefore may be underestimating the overall fluxes that
include the vegetation cover of the wetlands. Being able to measure extensively within the lakes systems provided evidence that the reed bed concentrations were far higher than that of the lakes themselves. Our data have shown significant need for increased recordings of diel cycles in all systems, with channels and lakes show significantly lower concentrations and fluxes when excluding these. Of our three water types, rivers had the smallest fluxes, showing that most of the $CH_4$ production must come from further within the wetlands. Most calculated $CH_4$ budgets, stem from extrapolations and data driven approaches
due to lack of process-based models (Saunois et al. 2020), therefore investigations of the interactions between reed stands and open water will be of high priority.

With our analysis of diel cycles both in the channels and the lakes we were able to further confirm the importance of adequate data collection, through 24-h coverage or specific correction for sampling bias, and implementation into models. The diel cycle



within the lake was consistent with stratification over the day, where vast amounts of organic carbon from macrophytes created

anoxic subsurface waters, which slowly and steadily mixed during the night. We showed that this cycle could have major consequences for spot measurements of concentrations and fluxes. Far larger quantities of $CH_4$ are released during the night due to daily stratification and with most current sampling techniques, such variability would be missed. A similar diel cycle was also active at the 'hot spot' site in a channel, where concentration changes varied four-fold between 4000-16000 nmol $L^{-1}$ indicating that the process of advective cooling during the night, should also be considered in shallow systems.

In summary, when comparing the overall peak-to-peak concentration ranges of observed diel cycles, there was a corresponding potential uncertainty of a factor of up to 4.5 within our measured lake (roughly 30%). Incurred by spot sampling without a dedicated sampling strategy taking diel variability into account. Using our measured examples with the diel cycles removed, accounted for an underestimation of up to 25% for channels, whereas an overestimation in lakes by 3.3% $CH_4$ concentration (nmol $L^{-1}$) (no diel cycles were measured in rivers). Including our measured diel cycle measurements, accounted for roughly

an increase of 20.4% in lakes and 4.2% decrease in channel fluxes. From this one study, this shows compelling evidence diel cycles must be accounted for when measuring concentrations and calculating fluxes and further proves, that the conventional picture of methane dynamics in freshwaters (Fig. 1) is too static. That further analysis of diel cycles must be included in the development of dynamic models of methane release from inland waters, especially with eutrophication predicted to. Given these cycles didn't just occur in lakes, a re-evaluation is needed on sampling techniques and data checks to include such cycles

from all water systems.

*Data availability.* All data presented in this paper are available from the corresponding author.





**Table A1.** Mean (± SD) for CH$_4$ concentration and flux (nmol L$^{-1}$ and $\mu$mol m$^{-2}$ h$^{-1}$ respectively) and O$_2$ concentration ($\mu$mol L$^{-1}$) across all measured lakes during each expedition (May, Aug or Oct). Grey shows the comparison with Pavel et al. (2009) which was measured in 2006 as a comparison for CH$_4$ concentration and flux, in the same units for May and September.

| | | Lake Uzlina | | | Lake Isac | | | Lake Puiu | | | Lake Roşu | | | Lake Roşuleţte | | |
|---|---|---|---|---|---|---|---|---|---|---|---|---|---|---|---|---|
| **Month** | | **May** | **Aug** | **Oct** | **May** | **Aug** | **Oct** | **May** | **Aug** | **Oct** | **May** | **Aug** | **Oct** | **May** | **Aug** | **Oct** |
| 2006 | | May | September | | May | September | | May | September | | May | September | | May | September | |
| **CH$_4$** nmol L$^{-1}$ | | **502 ± 81** | **2590 ± 391** | **722 ± 291** | **419 ± 62** | **1750 ± 827** | **539 ± 229** | **436 ± 301** | **n.d.** | **803 ± 635** | **468 ± 76** | **529 ± 297** | **756 ± 301** | **967 ± 1590** | **819 ± 426** | **515** |
| Pavel et al. (2009) | | 714 ± 257 | 644 ± 224 | | 569 ± 370 | 613 ± 161 | | 891 ± 424 | n.d. | | 534 ± 113 | 538 ± 119 | | 738 ± 575 | n.d. | |
| **CH$_4$ Flux** $\mu$mol L$^{-1}$ m$^{-2}$ h$^{-1}$ | | **13 ± 2.1** | **104 ± 15.4** | **17 ± 7.5** | **10.2 ± 1.7** | **82.6 ± 44.3** | **14.1 ± 6** | **11.9 ± 7.8** | **n.d.** | **17.8 ± 14.5** | **11.8 ± 2.5** | **19.3 ± 9.6** | **21.6 ± 11.2** | **21.7 ± 34.5** | **31.6 ± 18.1** | **29.3 ± 15.2** |
| Pavel et al. (2009) | | 18 ± 7 | 26 ± 9 | | 80 ± 51 | 62 ± 17 | | 23 ± 11 | n.d. | | 13 ± 3 | 7 ± 2 | | 11 ± 9 | n.d. | |
| **O$_2$** $\mu$mol L$^{-1}$ | | **267 ± 19** | **260 ± 8** | **325 ± 26.1** | **388 ± 39** | **294 ± 31.8** | **346 ± 16.3** | **280 ± 14.7** | **300 ± 39.1** | **75.3 ± 21.8** | **256 ± 13.5** | **255 ± 27.6** | **329 ± 11.4** | **218 ± 9.7** | **315 ± 3.6** | **311 ± 25** |





*Author contributions.* Anna Canning and Arne Körtzinger developed the concept and ideas of the study together. Anna Canning collected and processed the sensor data and wrote the manuscript. Arne Körtzinger and Bernhard Wehrli contributed ideas and clarifications of analyses. All authors reviewed and edited the manuscript.

*Competing interests.* All authors declare there is no conflict of interest.

*Acknowledgements.* The research leading to these results has received funding from the European Union's Horizon 2020 research and innovation program under the Marie Skłodowska-Curie grant agreement No 643052 (C-CASCADES project) and funding from Digital Earth which is coordinated by GEOMAR Helmholtz Centre for Ocean Research Kiel. We are grateful to Dennis Booge for assistance with the flux calculation. All of the -4H-JENA team with their support and help throughout my time with them and while on field work both day
and night; all those who assisted on the cruises: our Romanian colleagues captain Nice, George, Marian, Christian Teodoru and Marie-Sophie Maier who drove the campaigns





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
