# Peer review of "Methane in the Danube Delta: The importance of spatial patterns and diel cycles for atmospheric emission estimates"

_Biogeosciences, 2020_

## Referee Comment (RC1) · Anonymous Referee #1 · 12 Nov 2020

General comments

Overall, the study is an important contribution into methane dynamics in highly heterogenous system of river delta. It also emphasizes important (and often neglected) aspect of diel variability of CH4 dynamics. I see this study as an ambitious project as it aims to examine spatial and temporal variability of CH4 in various water bodies including river, side channels and lakes. It is important that authors distinguish between different aquatic systems and focus on more refined delta variability (river, channels, lakes) as these systems may experience distinct and highly variable CH4 dynamics. Also, this study not only aims to elucidate high spatial variability in the above subsystems of delta, but also adds additional layer of high temporal variability (diel) of CH4. Such study design allows to obtain more comprehensive picture of CH4 fluxes from such delta systems. In general, from reading the abstract and introduction I had an understanding of the study design and the rationale behind the investigations.

However, I see few substantial points which should be addressed before the manuscript can be published. These points are addressed in specific comments. Briefly, my major concerns are:

1. Gas transfer model used to calculate fluxes. As river delta is highly heterogenous system I believe that Cole and Caraco model is not the best choice to derive fluxes from measured concentrations. In the specific comments I propose the alternative gas transfer model, which could be more suitable in this system.

2. Although diel study is very important aspect here, extrapolation and recommendations based on just two full diel studies is certainly not enough to suggest how to design sampling campaign (time of samplings to reduce bias).

3. CO2 variability is certainly important and worth to study, however this manuscript focuses mostly on CH4. There is no information, which would introduce the reader the CO2 dynamics and in general CO2 appears in the text only fragmentary. I would suggest that spatial and temporal CO2 variability has a potential for separate manuscript.

4. There are numerous repetitions, especially regarding results. Values of concentrations and fluxes appear in tables and they are also described in the text, which is redundant.

5. Language is occasionally missing fluency and precision. Some sentences are too composite and reader gets lost, so their structure could be simplified to assure nicer reading of this interesting study.

Specific Comments

Abstract Overall abstract is understandable; however it is missing flow in reading oc-

casionally. Below I include several specific comments. However, if authors decide to implement other of my suggested comments (results and discussion) then the abstract would require modifications as well.

Line 7: please add how many days in total? "Over 3 seasons" is misleading and suggests that the concentrations were measured continuously for 3 seasons which is not the case.

Line 14-15: Did authors measured temperature profile to conclude there was a stratification? If not and this is an assumption (may be correct), it is not a result of the study

Line 15 – 16: Please correct this sentence, it doesn't have correct English structure.

Introduction:

I suggest to shorten introduction slightly. Especially selected parts (please see comments below) should be removed as they provide details which rather distract the reader from the main path of the manuscript instead of leading to the main issue. Instead I would suggest that authors elaborate slightly more on diel variability as this is main topic of the paper.

Line 29: It is not clear:, please specify "Potential driver" of what ?

Line 37 : substitute "although" with "yet"

Line 39-40: repetition "covering" and "covered", please change

Line 39: "give a range", just "range" is enough

Line 43-44: Do authors aim to assess the role of rivers and channels in methane emissions, not the role of methane?

Line 46-50: In my opinion this part is redundant as the methanogenesis or methanogenesis related processes are not a topic of this paper. According to me it deviates the

reader from the main (interesting) story.

Figure 1. This scheme is nice and illustrative but rather redundant in the context of this manuscript. This is rather text book knowledge and this scheme does not illustrate neither findings of the paper or concept of the study. In my opinion this scheme is not necessary.

Line 63: Please provide reference for this statement

Line 64 - 65: Although I see how browning or warming could affect seasonal variability of CH4, I cant see how they would impact diel variability?

Line 67-68: Real strength of this study: high spatial and temporal variability in different systems

Line 73: Please specify what is "extremely high resolution" 10 times a day, every hour, every minute...?

Methods:

Set up: As a reader not specifically familiar with this type of set up, I would appreciate more details. As far as I know this type of setup uses membrane-based equilibrators? I think this information should be in the method description so the reader obtains information about principle of the method. Also, equilibrators may underestimate concentrations (especially CH4), it would be highly appreciated if authors acknowledge drawbacks of this specific method. I do not try undermine this method and I am aware that every single method has its pros and cons, but for the reader it would be beneficial to know (very briefly) to know its drawback (and benefits too).

Study area is described sufficiently, however I am missing specific information how many km2 were covered regarding each system (lakes, channels, river)?

Line 109: Wind speeds may be locally different from what was occurring at the delta therefore its important how far from measurement sites Gorgova weather station was

located?

Line 110: I encourage authors to use reference Wannikhof 2014 instead of Wannikhof 1992

Line 113: I think its not correct to disregard stream velocities, especially without information about their values? What are the stream velocities (range) in this system?

Without this information it is difficult to assess if Cole and Caraco is the most suitable gas transfer model used. Please also see my comments below.

Line 114: Since authors didn't measure fluxes directly but derived them from the concentrations, it is very important to assure that authors apply gas transfer model which is the most suitable for measured sites and conditions. Cole and Caraco was developed for lakes mostly and doesn't include impact of variable hydrological settings in delta system, mainly flow velocity, which has an impact on fluxes too. Therefore, I would suggest to calculate fluxes using gas transfer model, including both variables (wind and flow velocity), which have an impact on k in such dynamic system. As there is no "ideal" model, I would still suggest to rather use the way k was derived in "Borges et al 2004" (as I see in reference list authors are already familiar with this publication) than Cole and Caraco gas transfer model.

Line 122: application of different model will have an impact on n value too

Results and discussion: As I already wrote in the General comments above, there are numerous places where results (values) are repeated. This is redundant. If value is already presented in the table, there is no need to repeat it in the text. It allows to keep the manuscript more concise. Also, regarding reporting values, I suggest to use mikromol L-1 instead of nmol L-1 all throughout the text.

Line 147: Please rephrase the sentence, oxygen by itself cant be undersaturated, lake water can be.

Line 152: "some variability ultimately due to temperature and wind": Currently I cant

see a support for this claim in the presented data. Where do the data show that flux variability was due to the temperature and wind?

Line 154: What do authors mean by "outgassing flux density"? Do you mean "mean flux"?

Line 154-156: If such extrapolation exercise is done, then it should rather be described in the method section. Thus, I suggest to move it to method section and in this section present only obtained results.

Also, I think it is important that authors measured and presented separate values for different types of water bodies in the delta. Its highly appreciated since such information is less common and shows that even it is one system, CH4 variability in water bodies located in close proximity to each other can be high. That is why I do not see the reason to pool all the data together and just derive one mean for the whole delta. To include flux variability within such complex system I suggest to (at least) calculate mean flux for each type of water body (channel, river, lake), calculate how much of area is occupied by each type of waters/wetlands and then extrapolate and derive total flux per year.

Line 162- 168: Good point, it is appreciated that authors acknowledge and discuss importance of ebullition in this system

Line 169 – 170: I am not familiar with specific set up of Maier et al 2020, but I do not agree that the method used by authors in the manuscript and chamber method cannot be compared at all. Sure, such high variability is missed by chambers, but at least fluxes at the same depths and same systems obtained by both studies could be compared.

Line 173: I agree that its not possible to assess impact of environmental drivers (such as wind) if it wasn't measured in situ. However, if authors suggest that distribution patterns were rather driven by biological or physical factors it should be supported by

the data. And I'm not sure if I see such evidence. Plus, which biological and physical factors do authors mean? It would be important to be more specific.

Line 181: Flooding is an important aspect in a frame of seasonality. As I imagine such system experiences reoccurring flooding, which may have big impact on CH4 fluxes (see for example Gatland et al 2014, JGR). To obtain more comprehensive picture on CH4 seasonality in delta system, it would be important to acknowledge flooding phenomena in a bit more detail.

Line 185-187: Please rewrite this sentence, I'm not sure what authors meant here.

Line 189-190: The sentence is structured like authors actually measured decomposition of OM and subsequent CH4 liberation. If authors didn't do such measurements, I would suggest to change it to:"....of biodegradation of OM could have triggered...which could be responsible for...."

Line 192: Please change to "may explain" or "could explain"

Line 193-194: Not only lakes are sources of labile organic matter, also rivers and channels too. Please provide reference

Line 195 - 196: "Channels......in O2": If authors make such statement, reference would be appreciated or these are results of this study? It is not clear.

Line 201: Why such conductivity points into groundwater influence?

Line 199 - 203 : Authors raise here the potential impact of groundwater on 'hot spot' conductivity. Also, the impact of groundwater on surface water CH4 concentrations may be very important too (especially in wetlands). So, if authors raise here importance of groundwater, then its potential impact on CH4 in this system should be acknowledged as well. Of course, one cant measure everything and I wouldn't expect the actual results but one or two sentences about potential impact of 'hot spot' groundwater on surface water CH4 in this system.

Line 202: Please clarify. What do authors mean by "water dropping further into the 'hot spot' ?

Line 206: In general the word "significant" is usually used in a statistical context, which is not the case here. I would suggest using other word

Line 211-222: I agree that this may be the case that more production happens in the delta, not in the river, and that such hot spots may impact the system's concentrations. Thus, this paragraph is important, however I have difficulty to follow the way it is currently presented, I guess because many values are included in the text. I would suggest maybe to draw simple conceptual figure, which could illustrate possible impact of channels and hot spot on a system. With colors representing concentrations? It would be easier to grab this interesting concept. If authors do not wish to illustrate this idea and stick to written text, I would suggest to rewrite it so its easier to follow.

Line 216- 229 Many values described in this paragraph are already in the table 1, so there is no need to write and describe all the values here again. It is unnecessary repetition and can be removed.

Situation of CH4 in lakes

I suggest changing the paragraph header to "CH4 concentrations and fluxes in lakes" or "CH4 dynamics in lakes" Many values described in this paragraph are already in the table 1, so there is no need to write and describe all the values here again. The discussion of obtained results is enough.

Line 254: What do authors mean by "averaging" the lakes? Is it average of fluxes, of total fluxes, from all lakes? Its unclear from the text.

Line 260: What authors mean by "intermediate" sampling ?

Diel CH4 cycling

Line 269 – 270: This sentence is not clear, requires clarification what authors intended

to say here?

Line 266: Please indicate the letters in Fig. 4 instead of "left"

Line 273-275: Do authors mean here convective mixing? If yes, its good to mention this term here

Line 284- 310 As I wrote in the general comments, CO2 deserves its own story. Including it suddenly into the manuscript distracts the reader from the main story, which as my understanding is, covers almost solely CH4. Also, so far the whole text above included only CH4 patterns. Additionally, this section has a title which points to CH4 daily patterns only. I suggest to remove CO2 dynamics from the manuscript.

Figure. 4: Nice and informative figures.

Line 326 – 327: I think authors mean here "day light data", not "day night data" ? Please clarify.

Line 334-354: It is a valid and very good point that authors' aim is to emphasize importance of diel variability and bias which occurs due to spot sampling. This may be one of the reasons for bias in current CH4 estimates or upscales. However, I would avoid giving recommendations how to minimize bias during lake CH4 sampling based on 2 full diel cycles. I see this manuscript as important work to bring attention into neglected issue of CH4 diel cycle in delta system, but to draw firm conclusions and give sampling recommendations, more full diel surveys would be necessary. Also, it is important to acknowledge that these two diel cycles captured by this study could be a snapshot as well. Thus, if being measured another day (or season), the opposite situation could have occurred with higher late daytime fluxes compared to the nigh time (for example, see Sieczko et al 2020 and references therein).

Table 2. I propose to visualize the table in a form of figure. It will be easier for the reader to see more clear the impact of day light (DL) vs. full diel cycle (FD) sampling on CH4 flux and concentration.

Line 349-350: The cited studies actually acknowledge and emphasize existence of CH4 diel cycle. They do not undermine it as this sentence suggests. The sentence or the references requires modification.

Line 379: Did authors measure organic carbon and showed that it was derived from macrophytes?

Technical Comments Line 170: please change "location" to "locations" Line 177: Please change "decease" to "decrease"

---

## Referee Comment (RC2) · Anonymous Referee #2 · 1 Dec 2020

This paper reports high-resolution measurements of dissolved CO2 and CH4, as well as O2, temperature and conductivity, in the Danube Delta. Three cruises were performed at different months of one year, covering both river, channel and lake systems. River Deltas are highly dynamic systems in terms of both biology and hydrology, and the present study is valuable in its purpose to quantify these dynamics in terms of greenhouse gas emission. This is an impressive dataset, consisting of highly resolved and high-quality measurements in this complex river delta, and I would really like to see it published.

Unfortunately, the paper has several shortcomings when it comes to presentation and

analysis. Several of these shortcomings are of rather fundamental nature. For example, even though rivers, channels and lakes are extensively discussed as being different from each with respect to greenhouse gases, there is no statistical analysis that supports the existence of any such differences. A "hot spot" of emission is mentioned without providing quantitative evidence for why it is different from the rest of sites. The variability of the gas exchange velocity can be massive at short scales, both in time and space, yet this is not accounted for, neither in calculations nor discussion. Diel cycles are presented and discussed in great length even though it really was only one measured diel cycle (the other diel cycle measurement was not fixed in space, and thus includes spatial variability); based on so little evidence, it seems not justified to draw far-reaching conclusions. There are also many issues with precision in writing, and evidence for many statements is lacking or unclear.

Nevertheless, the dataset of highly-resolved concentration measurements in this highly dynamic ecosystem seems robust, so with more effort, this could be turned into an interesting paper.

In revising this paper, I would like to urge the two senior co-authors to share their vast experience of writing papers with the junior first author.

**Detailed comments**

Title. Why only Methane? You also measured CO2, and that is worthwhile to report and communicate. Also, you only studied surface waters of the Danube Delta, and not the vast reed beds, which should be evident from the title.

L34. Source of what? Please specify.

L34. "Inland waters" are commonly defined as lakes, reservoirs and rivers. Wetlands are typically not part of inland waters.

L37. See the new lake CH4 emission estimate by Del Sontro et al. 2018, L&O.

L41. This sentence is repetitive.
L43. To my knowledge, there is no definition of "channel"; aren't these just running waters that are somehow anthropogenically modified, such as many rivers and streams?

L47. "end of line respiration process" sounds like colloquial language, please revise.

L48-50. No need to go into pathways of methane production, since it is not at all part of this paper. Also Figure 1 is not really needed, since this study is not about establishing a lake methane budget.

L60-64. Unclear in how far this is relevant introduction to this paper. Instead, focus the introduction on "spatial and temporal variability" (L64), because that's what this paper is about.

L66. Unclear what "monitoring approaches tend to stay within one system" means.

L76. Objective 2 should be rephrased, since you do not address global-scale fluxes in this paper, and you only measured one diel cycle.

Section 2.2. Please describe what distinguishes rivers from channels.

Figure 2. This map shows the travelled track (how long was it in total?), but couldn't you also make maps that show the concentrations of CO2 and CH4 along this track? This would be a very intuitive way to visualize the data. Also: the yellow lake complexes were not studied and do not need to be highlighted. And instead of the various denominations (3, 4, b1, b2, i-v), what about writing the respective names onto the map?

Figure 2 caption: what does "with only slight variations" mean?

L99-100- Is this the annual temperature range, or daily? And what makes it extreme?

L102. "thorough" is subjective, and can be skipped.

L120-125. I have some serious concerns with the way the gas exchange velocity and flux estimates were treated and discussed. Only the concentrations CH4water and

BGD
CO2water were measured. The concentrations in air were assumed to be at global average, which is doubtful in such a biologically active wetland area. The gas exchange velocity k was scaled from wind speed (unclear where it was measured) for both lakes and rivers (albeit different exponents were used for lakes and rivers). I assume that the channels were treated as the rivers? My point is that k is highly variable at short time scales, and largely driven by hydrodynamics, which in turn varies with wind speed, but also with water flow, hydromorphology, and thermal structure. These sources of variability will vary in time and space and between types of systems, and are very unlikely to be captured by scaling from wind speed. The authors need to acknowledge that, and add some discussion on the reliability of their estimate of k. Given that apparently a study of greenhouse gas emission from the Danube Delta using floating chambers was published recently (Maier et al.), the authors could use the measured k values from that study for calculation of their own fluxes, or to assess in how far the wind speedscaled k values are congruent with measurements of k. The robust treasure of this study are the highly-resolved and repeated concentration measurements, and it needs to be made clearer that the fluxes reported here are estimates, not measurements. Another important aspect of equation 3: the gas exchange velocity and concentration influence each other. A very high k can quickly empty the water of gases and thus lead to low concentrations, and low k prevents emission and can lead to the build-up of high concentrations. It may therefore very well be possible that the sites where the authors have observed high concentrations, the fluxes may not be high if that site was characterized by very stagnant water (possibly in the "hot spot" channel?), instead concentrations might have been high because k and thus flux was low. A relevant paper on the spatial decoupling of k, concentration and emission is Rocher-Ros et al. 2019, L&O, their Figs. 2 and 3. This aspect should be added to the discussion.

L126. This statement needs a reference.

Results and Discussion: I wonder if it would not be helpful to separate the Results from the Discussion, and to present the results step by step (concentrations, maps of
concentration, then estimates of emission flux, then an upscaled emission for the entire Delta), to then Discuss the ensemble of the findings.

L132. Consistent, not constant.

L133. Another point of serious concern: This study completely lacks statistical testing of the reported differences, and the term "significant" should only be used if a statistical test can support that the difference between e.g. systems or sampling campaigns was statistically significant. The authors must include statistical testing in their revision.

L134. Using maximum values is not very helpful, better to report means, medians or ranges.

L137. What is meant by "water type boundaries"?

- L138. If I remember right, Crawford et al. studied streams, not channels.
- L139. "were found to have higher concentrations" where is this visible?
- L140. What is a "boundary crossover"?

L142. On the map, there is channel north of Lake Puiu?

L143. Using the term "hot spot anomaly" requires some quantitative and statistical underpinning. It seems from Table 1 that this site was only showing elevated CH4 in Aug, but not in May and Oct. So is this site really significantly different from other sites, i.e. other river reaches, or other lakes, or other channels? Statistical testing is warranted.

L146. Briefly explain that in October, macrophytes senesce and can be expected to start decomposing in the water.

L147. "measurements were not distributed proportionally" – this not only applies to O2 measurements, but to all measurements, so this would affect all your data and conclusions?

BGD
L150. This sentence is speculative and should be removed.

Figure 3. I suggest to present only concentrations, and give some aggregated numbers for fluxes later. Fluxes are only calculated estimates, which are derived from your actual measurements. Instead, also include CO2 concentrations here. And please include statistical testing to infer any differences between categories. Also, please label the panels of this figure. Two observations: O2 saturation was frequently very low, indicating strong respiration in the water or the reed belt. And the distribution of CH4 was very skewed, with generally rather low values, but quite a bunch of very high values.

L152. Fluxes correspond to concentrations because your k estimate is essentially a constant, which k certainly is not in nature. On the contrary, it can be very variable at short scale of space and time. This observation is an artefact.

L154. For upscaling, it is very important to detail how the calculations were performed, and which assumptions were made, step by step. This was not really the case here.

L159 "this estimate " - which estimate?

L161-164. Confusing that both a 277% and 70% underestimation of total flux are cited. Using the 70% estimate seems more realistic, because that stems from the same system.

L169-170. Use those floating chamber measurements to calculate k values, which you then can use for your upscaling. It would also be informative to compare the floating chamber measurements of emission to your calculations of emission.

Table 1. These are descriptive statistics. Also, please include CO2 here and save CH4 flux for later. "stinky channel hot spot" does not seem appropriate terminology, and it does also not seem to have extremely high concentrations compared to the other channels. And what does the footnote \*\* mean?

L173. Please show these correlations, or give regression statistics in the text. This
sentence could also be interpreted as an indication that scaling k from wind speed at some met station was not really relevant.

L175-176. Which external factors? Aren't the most important factors biological and physical?

L179. Change with respect to what? And again, "significantly" requires some form of statistical testing.

L183. Concentrations and stuartion of what?

L188. I would expect that macrophyte degradation should also be high in the channels, not only the lakes?

L189. This could be explored further. With your data, you could make maps and actually at which locations concentration were elevated. For the lakes, you might want to make a correlation between distance from shore and concentration.

L194. Methanogenesis takes places in anoxic sediments, and I assume the channels don't have very much sediment accumulation at their bottoms?

Section 3.1.3. Again, this needs to statistically supported.

L209. Movement of water?

Section 3.1.4. Unclear what "fluvial" is. Everything minus lakes? Are channels included? And aren't the lakes part of the fluvial delta?

L213. "Little evidence" – please show the evidence that you have.

L222. This is expected, since there is very little sediment accumulation expected at the bottom of rivers.

L230. Unclear what time period this estimate covers. The three months of measurement? Or the entire year, based on the 3 sampling occasions.

L242. This is evidence that the emission might not have changed much, but for as-
sessing eutrophication, you would need data on phosphorus, nitrogen or chlorophyll.

L246-249. Here's several statements that require to be supported by showing evidence: enhanced CH4 production, increasing concentrations coming to the lakes, oxidation, visible on the edges of lakes.

L250-251. Which changes in morphology? What evidence is there for higher productivity in the channels. And what is meant by "macrophyte distributions"?

L256. Ebullition is also a flux from the sediment.

L260. With your data, you are in a very good position to explore local dynamics, by making maps and showing them.

Section 3.2. This section is far too long, mainly because there was only one true diel cycle measured; during the other diel-cycle measurements, the boat was moving, and thus spatial variability is included in the measurement. Also, the authors lack data that help to explain the diel cycle, e.g. water column profiles of temperature (to address convection) or of gases, measurements of k, or similar. Therefore, the discussion is quite vague. Based on so little data and machnistic understanding, it does not seem warranted to draw the conclusion that diel cycles are important in the Danube Delta, and need to be accounted for (e.g. in the abstract, or L339-341)

Figure 4. I would prefer simpler plots, with time of the x axis and the analytes on the y axis.

Figure 5. I would like to see more of this! More maps with concentrations, and further analyses of spatial patterns of elevated (and low) concentrations.

L370. Is there any data or other evidence for high concentrations in the reed bed?

BGD

---

## Author Response (AR1)

**Responses to Anonymous Referee 1**

**General comments**

Overall, the study is an important contribution into methane dynamics in highly heterogenous system of river delta. It also emphasizes important (and often neglected) aspect of diel variability of $CH_4$ dynamics. I see this study as an ambitious project as it aims to examine spatial and temporal variability of $CH_4$ in various water bodies including river, side channels and lakes. It is important that authors distinguish between different aquatic systems and focus on more refined delta variability (river, channels, lakes) as these systems may experience distinct and highly variable $CH_4$ dynamics. Also, this study not only aims to elucidate high spatial variability in the above subsystems of delta, but also adds additional layer of high temporal variability (diel) of $CH_4$.

Such study design allows to obtain more comprehensive picture of $CH_4$ fluxes from such delta systems. In general, from reading the abstract and introduction I had an understanding of the study design and the rationale behind the investigations.

**Response: Thank you for your positive assessment. Definition of the areas will be made clearer and each comment below has been individually commented to.**

However, I see few substantial points which should be addressed before the manuscript can be published. These points are addressed in specific comments. Briefly, my major concerns are:

**1.** Gas transfer model used to calculate fluxes. As river delta is highly heterogenous system I believe that Cole and Caraco model is not the best choice to derive fluxes from measured concentrations. In the specific comments I propose the alternative gas transfer model, which could be more suitable in this system.

**Response: See specific response to Line 144**

**2.** Although diel study is very important aspect here, extrapolation and recommendations based on just two full diel studies is certainly not enough to suggest how to design sampling campaign (time of samplings to reduce bias).

**Response: We agree and we will remove explicit recommendations. The extrapolations merely showed the potential bias, and we indicate multiple times that the calculations were highly specific to our observation (e.g. line 337). In the revised version we will communicate this more carefully.**

**3.** $CO_2$ variability is certainly important and worth to study, however this manuscript focuses mostly on $CH_4$ There is no information, which would introduce the reader the $CO_2$ dynamics and in general $CO_2$ appears in the text only fragmentary. I would suggest that spatial and temporal $CO_2$ variability has a potential for separate manuscript.

**Response: This is a constructive suggestion and we will remove the $CO_2$ data in the revised manuscript to make the story more concise.**

**4.** There are numerous repetitions, especially regarding results. Values of concentrations and fluxes appear in tables and they are also described in the text, which is redundant.

**Response: The text, figures and tables will be revised accordingly to avoid repetitions, thank you.**

**5.** Language is occasionally missing fluency and precision. Some sentences are too composite and reader gets lost, so their structure could be simplified to assure nicer reading of this interesting study.

**Response: We will edit language and overall structure for clarity in order to make the m.s. more accessible for the readers.**

**Specific Comments (labelled SC#)**

Overall abstract is understandable; however it is missing flow in reading occasionally. Below I include several specific comments. However, if authors decide to implement other of my suggested comments (results and discussion) then the abstract would require modifications as well.

**Response: Thank you, we will rewrite the abstract to reflect changes made and to improve the overall flow.**

**SC1.** Line 7: please add how many days in total? "Over 3 seasons" is misleading and suggests that the concentrations were measured continuously for 3 seasons which is not the case.

**Response: The sentence will read "During three expeditions in different seasons..."**

**SC2.** Line 14-15: Did authors measured temperature profile to conclude there was a stratification? If not and this is an assumption (may be correct), it is not a result of the study

**Response: The statement is backed up by temperature measurements.**

**SC3.** Line 15 – 16: Please correct this sentence, it doesn't have correct English structure.

**Response: The edited sentence now reads: "Day-time spot sampling techniques would miss the effect of diel cycles and underestimate average methane concentrations by 25 % for channels"**

**Introduction:**

I suggest to shorten introduction slightly. Especially selected parts (please see comments below) should be removed as they provide details which rather distract the reader from the main path of the manuscript instead of leading to the main issue. Instead I would suggest that authors elaborate slightly more on diel variability as this is main topic of the paper.

**Response: We will shorten the introduction and emphasize the diel variability.**

**SC4.** Line 29: It is not clear:, please specify "Potential driver" of what ?

**Response: The modified sentence will read "Biogenic emissions from wetlands (Nisbet et al. 2019) contribute strongly to the overall estimate of 159 (117-212) Tg $CH_4$ $yr^{-1}$ from inland waters ..."**

**SC5.** Line 37 : substitute "although" with "yet"

**Response: This will be implemented.**

**SC6.** Line 39-40: repetition "covering" and "covered", please change

**Response: This will be changed.**

**SC7.** Line 39: "give a range", just "range" is enough

**Response: This will be changed.**

**SC8.** Line 43-44: Do authors aim to assess the role of rivers and channels in methane emissions, not the role of methane?

**Response: The edited sentence will read: "Therefore, there is a need for more detailed assessment of the role of rivers and channels for methane emissions, as they have been suggested to be more spatiotemporally variable for $CH_4$ than $CO_2$"**

**SC9.** Line 46-50: In my opinion this part is redundant as the methanogenesis or methanogenesis related processes are not a topic of this paper. According to me it deviates the reader from the main (interesting) story.

**Response: These lines will be removed to make the story more concise.**

**SC10.** Figure 1. This scheme is nice and illustrative but rather redundant in the context of this manuscript. This is rather text book knowledge and this scheme does not illustrate neither findings of the paper or concept of the study. In my opinion this scheme is not necessary.

**Response: Figure 1 will be removed in the revised version.**

**SC11.** Line 63: Please provide reference for this statement

**Response: We refer to the reference (Bartosiewicz et al., 2019) in the sentence before. We will make this clear: "These authors suggest that $CH_4$ production in bottom waters may increase, potentially leading to . . . "**

**SC12.** Line 64 - 65: Although I see how browning or warming could affect seasonal variability of CH4, I cant see how they would impact diel variability?

**Response: Browning is as a complex driver for methane emissions mostly at high latitudes, but as this study has a different focus we decided to omit a reference to it. New line reads: "suggest that global warming will increase surface water temperatures and strengthen lake stratification (Woolway et al., 2019 )".**

**SC13.** Line 67-68: Real strength of this study: high spatial and temporal variability in different systems

**Response: We will emphasize this point as follows: "The acquired high spatial and temporal resolution of methane concentrations and corresponding emissions formed a unique observational data basis. Continuous measurements across the delta allowed us to assess the importance of different systems (lakes, rivers and channels) and the high-frequency data at specific sites yielded insights into diel cycles and the specific day-night dynamics of methane emissions."**

**SC14.** Line 73: Please specify what is "extremely high resolution" 10 times a day, every hour, every minute: : :?

**Response: Up to one measurement per second. "Here, we take a complementary approach with a measurement frequency up to 1 Hz. This allows not only for high-resolution data both in time and space but also for a detailed look at the diel variability time-scale"**

**Methods:**

**SC15.** Set up: As a reader not specifically familiar with this type of set up, I would appreciate more details. As far as I know this type of setup uses membrane-based equilibrators? I think this information should be in the method description so the reader obtains information about principle of the method. Also, equlibrators may underestimate concentrations (especially $CH_4$), it would be highly appreciated if authors acknowledge drawbacks of this specific method. I do not try undermine this method and I am aware that every single method has its pros and cons, but for the reader it would be beneficial to know (very briefly) to know its drawback (and benefits too).

**SC16.** Study area is described sufficiently, however I am missing specific information how many km2 were covered regarding each system (lakes, channels, river)?

**Response: In order keep the text concise and avoid repetition of what has already been published, we refer to the technical note Canning et al. (2020).**

**SC17.** Line 109: Wind speeds may be locally different from what was occurring at the delta therefore its important how far from measurement sites Gorgova weather station was located?

**Response: Gorgova is roughly in the middle of the delta: "Barometric pressure as well as wind speed measured at the Gorgova station in the center of the delta were used".**

**SC18.** Line 110: I encourage authors to use reference Wannikhof 2014 instead of Wannikhof 1992

**Response: We will change the reference as requested. Based on a quick recalculation, however, the switch to Wanninkhof 2014 will not significantly affect the results; the difference will be smaller than the measurement error for concentrations.**

Line 113: I think its not correct to disregard stream velocities, especially without information about their values? What are the stream velocities (range) in this system?

**Response: Stream velocities within the delta were slow with maxima smaller than 30 cm s$^{-1}$. With the exception of flood events, the hydrodynamics of smaller canals is much closer to wetland lakes than to estuaries for which Borges et al. (2004) derived their k values. We therefore insist that the Cole and Caraco model is more adequate than other parametrizations for river systems.**

Without this information it is difficult to assess if Cole and Caraco is the most suitable gas transfer model used. Please also see my comments below.

**SC19.** Line 114: Since authors didn't measure fluxes directly but derived them from the concentrations, it is very important to assure that authors apply gas transfer model which is the most suitable for measured sites and conditions. Cole and Caraco was developed for lakes mostly and doesn't include impact of variable hydrological settings in delta system, mainly flow velocity, which has an impact on fluxes too. Therefore, I would suggest to calculate fluxes using gas transfer model, including both variables (wind and flow velocity), which have an impact on k in such dynamic system. As there is no "ideal" model, I would still suggest to rather use the way k was derived in "Borges et al 2004" (as I see in reference list authors are already familiar with this publication) than Cole and Caraco gas transfer model.

**Response: Indeed, there is no ideal model for gas exchange in slowly moving fluvial systems such as the channels of the Danube Delta. For a river with flow velocities that were typically ten times faster than those observed in the Danube Delta DelSontro et al. (2016) calculated the gas transfer coefficient with nine model equations resulting in a range of k**

values spanning an order of magnitude. For quasi-stagnant waters, which we observed in the delta, however, the lake Cole and Caraco is quite well established. By contrast, Borges et al. (2004) worked on large rivers and a fjord, systems that are hydrodynamically very different from low-flow regime of the lakes and channels of the delta. In support of our approach, we will follow the approach of Cole et al. (2010) and extend the comparison with the floating chamber measurements by Maier et al. (2020) – see lines 167 169.

**SC20.** Line 122: application of different model will have an impact on n value too

**Response: See above.**

**Results and discussion:**

**SC21.** As I already wrote in the General comments above, there are numerous places where results (values) are repeated. This is redundant. If value is already presented in the table, there is no need to repeat it in the text. It allows to keep the manuscript more concise. Also, regarding reporting values, I suggest to use $\mu$mol L$^{-1}$ instead of nmol L$^{-1}$ all throughout the text.

**Response: This will edit the results section accordingly and use $\mu$mol L$^{-1}$ throughout.**

**SC22.** Line 147: Please rephrase the sentence, oxygen by itself cant be undersaturated, lake water can be.

**Response: "Oxygen concentration in the water was mostly below saturation…"**

**SC23.** Line 152: "some variability ultimately due to temperature and wind": Currently I cant see a support for this claim in the presented data. Where do the data show that flux variability was due to the temperature and wind?

**Response: This was a general statement and will be removed.**

**SC24.** Line 154: What do authors mean by "outgassing flux density"? Do you mean "mean flux"?

**Response: Yes, corrected as follows "we got an overall mean outgassing flux of 49 $\pm$ 61 $\mu$mol m$^{-2}$ h$^{-1}$"**

**SC25.** Line 154-156: If such extrapolation exercise is done, then it should rather be described in the method section. Thus, I suggest to move it to method section and in this section present only obtained results. Also, I think it is important that authors measured and presented separate values for different types of water bodies in the delta. Its highly appreciated since such information is less common and shows that even it is one system, CH$_4$ variability in water bodies located in close proximity to each other can be high. That is why I do not see the reason to pool all the data together and just derive one mean for the whole delta. To include flux variability within such complex system I suggest to (at least) calculate mean flux for each type of water body (channel, river, lake), calculate how much of area is occupied by each type of waters/wetlands and then extrapolate and derive total flux per year.

**Response: This is what we did. We took the areas of each water body (lake, rivers and channels) and to these applied the average fluxes observed for the three water types. (lines 152 - 160). The concentrations and fluxes are detailed in Table 1. We will add a sentence outlining the upscaling per system. "We used the estimated area from Maier et al. (2020) for total area of rivers, channels and lakes (164, 33, 258 km$^2$ respectively) and the average emission rates in Table 1. Taking the average across all seasons, annual estimates for methane emissions of 16.1, 81.9 and 24.9 $\mu$mol m$^{-2}$ h$^{-1}$, for rivers, channels and lakes, respectively. The combined overall mean outgassing flux is then 49 $\pm$ 61 $\mu$mol m$^{-2}$ h$^{-1}$."**

**SC26.** Line 162- 168: Good point, it is appreciated that authors acknowledge and discuss importance of ebullition in this system

**Response: Thank you!**

**SC27.** Line 169 – 170: I am not familiar with specific set up of Maier et al 2020, but I do not agree that the method used by authors in the manuscript and chamber method cannot be compared at all. Sure, such high variability is missed by chambers, but at least fluxes at the same depths and same systems obtained by both studies could be compared.

**Response: Chamber measurements are picking up ebullition, which can be high, whereas our estimates for diffuse fluxes were based on concentration measurements in the surface waters. The two methods are therefore influenced by different processes. However, some general comparison is possible and will be implemented.**

**SC28.** Line 173: I agree that its not possible to assess impact of environmental drivers (such as wind) if it wasn't measured in situ. However, if authors suggest that distribution patterns were rather driven by biological or physical factors it should be supported by the data. And I'm not sure if I see such evidence. Plus, which biological and physical factors do authors mean? It would be important to be more specific.

**Response: This paragraph with reference to biological and physical effects comes too early and we will delete it. It will be more evident, when we discuss the patterns shown in Figure 5.**

**SC29.** Line 181: Flooding is an important aspect in a frame of seasonality. As I imagine such system experiences reoccurring flooding, which may have big impact on CH4 fluxes (see for example Gatland et al 2014, JGR). To obtain more comprehensive picture on CH4 seasonality in delta system, it would be important to acknowledge flooding phenomena in a bit more detail.

**Response: Good point we will include a reference to flooding and flood recession. Flooding will push oxygenated water into the reed stands and decrease emissions, while flood recession will move anoxic water from the reed into the channels and trigger ebullition.**

**SC30.** Line 185-187: Please rewrite this sentence, I'm not sure what authors meant here.

**Response: Rephrased to, 'Aug had the lowest water levels of each season, and although it showed the largest $CH_4$ range among the seasons, it had the lowest measured median values, coinciding with the hypothesis that there is an overall decreased $CH_4$ concentration values during lower water levels (Melack et al. 2004; Marín-Muñiz et al. 2015; McGinnis et al. 2016).**

Line 189-190: The sentence is structured like authors actually measured decomposition of OM and subsequent CH4 liberation. If authors didn't do such measurements, I would suggest to change it to:": : :.of biodegradation of OM could have triggered: : :which could be responsible for: : :."

**Response: This will be changed thank you!**

**SC31.** Line 192: Please change to "may explain" or "could explain"

**Response: Will be changed!**

**SC32.** Line 193-194: Not only lakes are sources of labile organic matter, also rivers and channels too. Please provide reference

**Response: Good point "River reaches, channels and lakes are sources of labile organic carbon that fuels methanogenesis (Schubert & Wehrli , 2019).**

**SC33.** Line 195 - 196: "Channels..: : :..in O2": If authors make such statement, reference would be appreciated or these are results of this study? It is not clear.

**Response: Will insert reference to Maier et al. 2020, where $O_2$ data for all three systems have been evaluated.**

**SC34.** Line 201: Why such conductivity points into groundwater influence?

**Response: High conductivity is a reliable tracer for groundwater exfiltration (see Harvey et al., 1997). We found this region had a peak in conductivity compared to regions surrounding this location.**

**SC35.** Line 199 - 203 : Authors raise here the potential impact of groundwater on 'hot spot' conductivity. Also, the impact of groundwater on surface water $CH_4$ concentrations may be very important too (especially in wetlands). So, if authors raise here importance of groundwater, then its potential impact on $CH_4$ in this system should be acknowledged as well. Of course, one cant measure everything and I wouldn't expect the actual results but one or two sentences about potential impact of 'hot spot' groundwater on surface water $CH_4$ in this system.

**Response: Groundwater can have an impact on overall gas supersaturation within the water column (Crawford et al., 2014), potentially leading to increased $CH_4$ concentrations within specific locations throughout the delta.**

**SC36.** Line 202: Please clarify. What do authors mean by "water dropping further into the 'hot spot' ?

**Response: Water temperature decreased the further away we got from the channels joining to the hot spot. Rephrased to, 'Given the dramatic change within the concentrations and properties of the water, such as the water temperature decreasing the further away from the channel we travelled into the 'hot spot', even within summer, this would further provide evidence from cooler groundwaters or potential waters from the reed beds also suggested by Maier et al. (2020).'**

**SC37.** Line 206: In general the word "significant" is usually used in a statistical context, which is not the case here. I would suggest using other word

**Response: We replace "significant" by "strong" influence**

**SC38.** Line 211-222: I agree that this may be the case that more production happens in the delta, not in the river, and that such hot spots may impact the system's concentrations. Thus, this paragraph is important, however I have difficulty to follow the way it is currently presented, I guess because many values are included in the text. I would suggest maybe to draw simple conceptual figure, which could illustrate possible impact of channels and hot spot on a system. With colors representing concentrations? It would be easier to grab this interesting concept. If authors do not wish to illustrate this idea and stick to written text, I would suggest to rewrite it so its easier to follow.

**Response: We prefer changing the text as follows: "The fluvial delta (rivers and channels) works as the supply of incoming water into the main part of the delta, accounting for the base level of $CH_4$ concentrations being laterally transported. We found very little evidence that intrusions from the Black Sea may have reached into the delta and have an impact such as suggested before (Durisch-Kaiser et al. 2008; Pavel et al. 2009). This would be important to explain reduced methane production as sulfate reduction becomes the dominating anaerobic mineralization pathway. Rivers had the lowest range of concentrations for $CH_4$ with the smallest variability out of all systems and the delta (Fig. 3). When excluding the 'hot spot', median values for channels were larger than those for rivers and fairly consistent throughout May and Aug while increasing during Oct. While in comparison, the hot spot measured the largest concen-**

trations during May and Aug respectively, and thereby changed the overall channel dynamics during Aug by increasing the overall channel median. The influence of the hot spot showed the significant influence one spot can have on a system, providing evidence that most of the $CH_4$ production happens within the delta, not the river itself.'

**SC39.** Line 216- 229 Many values described in this paragraph are already in the table 1, so there is no need to write and describe all the values here again. It is unnecessary repetition and can be removed.

**SC40.** I suggest changing the paragraph header to "CH4 concentrations and fluxes in lakes" or "CH4 dynamics in lakes" Many values described in this paragraph are already in the table 1, so there is no need to write and describe all the values here again. The discussion of obtained results is enough.

**Response: We will edit the results section and delete redundant values.**

**SC41.** Line 254: What do authors mean by "averaging" the lakes? Is it average of fluxes, of total fluxes, from all lakes? Its unclear from the text.

**Response: Average fluxes over all of the measured lakes. Will be made clearer.**

**SC42.** Line 260: What authors mean by "intermediate" sampling ?

**Response: Intermediate is meant as sampling at specific times, or in specific places such as just at channels joining the lakes. This will be made clearer.**

**Diel CH4 cycling**

**SC43.** Line 269 – 270: This sentence is not clear, requires clarification what authors intended to say here?

**Response: Rephrased to, ' As the mapping transect in Lake Rosu started around 9:00, some spatial variability from varying concentrations due to proximity to the shore line (Fig. 5) is superimposed onto the dominant diel cycle, causing $CH_4$ concentrations to vary over the range 200–500 nmol $L^{-1}$.**

**SC44.** Line 266: Please indicate the letters in Fig. 4 instead of "left"

**Response: This will be implemented**

**SC45.** Line 273-275: Do authors mean here convective mixing? If yes, its good to mention this term here

**Response: Corrected: "A possible explanation for this hysteresis: the water column stratifies during the day, and undergoes convective mixing as the surface water is cooling during the night"**

**SC46.** Line 284- 310 As I wrote in the general comments, $CO_2$ deserves its own story. Including it suddenly into the manuscript distracts the reader from the main story, which as my understanding is, covers almost solely $CH_4$. Also, so far the whole text above included only $CH_4$ patterns. Additionally, this section has a title which points to $CH_4$ daily patterns only. I suggest to remove $CO_2$ dynamics from the manuscript.

**Response: $CO_2$ will be removed from this manuscript.**

**SC47.** Figure. 4: Nice and informative figures.

**Response: Thank you!**

**SC48.** Line 326 – 327: I think authors mean here "day light data", not "day night data" ? Please clarify.

**Response: Yes, will be corrected.**

**SC49.** Line 334-354: It is a valid and very good point that authors' aim is to emphasize importance of diel variability and bias which occurs due to spot sampling. This may be one of the reasons for bias in current $CH_4$ estimates or upscales. However, I would avoid giving recommendations how to minimize bias during lake $CH_4$ sampling based on 2 full diel cycles. I see this manuscript as important work to bring attention into neglected issue of $CH_4$ diel cycle in delta system, but to draw firm conclusions and give sampling recommendations, more full diel surveys would be necessary. Also, it is important to acknowledge that these two diel cycles captured by this study could be a snapshot as well. Thus, if being measured another day (or season), the opposite situation could have occurred with higher late daytime fluxes compared to the nigh time (for example, see Sieczko et al 2020 and references therein).

**Response: We will remove the recommendations and focus more on the spatial variability as previously noted. Acknowledgement of few diel cycles and potentially only capturing a 'snap shot' will be implemented, 'Although capturing the diel variability, it must be noted that few diel cycles were captured and these may well be different at other times and locations and therefore not be representative of the overall situation in the delta.'**

**SC50.** Table 2. I propose to visualize the table in a form of figure. It will be easier for the reader to see more clear the impact of day light (DL) vs. full diel cycle (FD) sampling on $CH_4$ flux and concentration.

**Response: We will add such a figure as suggested.**

**SC51.** Line 349-350: The cited studies actually acknowledge and emphasize existence of $CH_4$ diel cycle. They do not undermine it as this sentence suggests. The sentence or the references requires modification.

**Response: This sentence was meant to reference said references as acknowledging the existence of diel cycles. Therefore, to make this clearer, this will be modified to: 'There have been multiple studies looking into diel cycles (see Nimick et al. 2011; Zhang et al. 2018; van Bergen et al. 2019; 350 Sieczko et al. 2020 for examples), yet these are usually undetected or not fully resolved and therefore ignored, particularly in studies with...'**

**SC52.** Line 379: Did authors measure organic carbon and showed that it was derived from macrophytes?

**Response: Unfortunately, this was not measured by us.**

**SC53.** Technical Comments Line 170: please change "location" to "locations"

**Response: done**

**SC54.** Line 177: Please change "decease" to "decrease"

**Response: done**

**Responses to Anonymous Referee 2**

This paper reports high-resolution measurements of dissolved $CO_2$ and $CH_4$, as well as $O_2$, temperature and conductivity, in the Danube Delta. Three cruises were performed at different months of one year, covering both river, channel and lake systems. River Deltas are highly dynamic systems in terms of both biology and hydrology, and the present study is valuable in its purpose to quantify these dynamics in terms of greenhouse gas emission. This is an impressive dataset, consisting of highly resolved and high-quality measurements in this complex river delta, and I would really like to see it published.

**Response: Thank you for your positive comments! We have replied to each one individually below and the review helps to improve the manuscript significantly.**

Unfortunately, the paper has several shortcomings when it comes to presentation and analysis. Several of these shortcomings are of rather fundamental nature. For example, even though rivers, channels and lakes are extensively discussed as being different from each with respect to greenhouse gases, there is no statistical analysis that supports the existence of any such differences. A "hot spot" of emission is mentioned without providing quantitative evidence for why it is different from the rest of sites. The variability of the gas exchange velocity can be massive at short scales, both in time and space, yet this is not accounted for, neither in calculations nor discussion. Diel cycles are presented and discussed in great length even though it really was only one measured diel cycle (the other diel cycle measurement was not fixed in space, and thus includes spatial variability); based on so little evidence, it seems not justified to draw far-reaching conclusions. There are also many issues with precision in writing, and evidence for many statements is lacking or unclear.

**Response: In the revised version we will strive for a more careful balance and avoid too far-reaching conclusions. Details are explained below.**

Nevertheless, the dataset of highly-resolved concentration measurements in this highly dynamic ecosystem seems robust, so with more effort, this could be turned into an interesting paper.

In revising this paper, I would like to urge the two senior co-authors to share their vast experience of writing papers with the junior first author.

**Detailed comments**

**SC1.** Title. Why only Methane? You also measured $CO_2$, and that is worthwhile to report and communicate. Also, you only studied surface waters of the Danube Delta, and not the vast reed beds, which should be evident from the title.

**Response: We will follow the suggestions by the other reviewer, and remove the $CO_2$ data to focus on a more concise a story $CH_4$ concentrations and fluxes. Surface waters will be added to the title, working title of: 'Surface water Methane in the Danube Delta: The importance of spatial patterns and diel cycles for atmospheric emission estimates.' The $CO_2$ data will be dealt with elsewhere.**

**SC2.** L34. Source of what? Please specify.

**Response: Source of $CH_4$. New sentence in next comment below.**

**SC3.** L34. "Inland waters" are commonly defined as lakes, reservoirs and rivers. Wetlands are typically not part of inland waters.

**Response: Rephrased as "Natural wetlands are one of the single largest sources of methane (125–218 Tg CH$_4$ yr$^{-1}$), accounting for roughly one third of total (anthropogenic and natural) emissions (Dean et al. 2018; Saunois et al. 2019)"**

**SC4.** L37. See the new lake CH4 emission estimate by Del Sontro et al. 2018, L&O.

**Response: We replace Krischke et al. by DelSontro et al. 2018 as follows: "DelSontro et al. (2018) illustrate the large uncertainties in these data. Depending on the upscaling methods these authors arrive at global CH4 emission rates from lakes in the range of 78 – 248 Tg CH$_4$ C yr$^{-1}$."**

**SC5.** L41. This sentence is repetitive.

**Response: We reorganize this section, starting with lakes and ending with rivers to avoid repetition:**

**Wetlands are one of the single largest source within the inland waters (125–218 Tg CH$_4$ yr$^{-1}$) accounting for roughly one third of total emissions (Dean et al. 2018; Saunois et al. 2019). They are usual intertwinted with rivers, channels and lakes making them highly diverse regions. Due to lakes being some of the easier systems to measure and compare, they are the most extensively covered components of inland waters although only covering 0.9% of the Earth's surface. DelSontro et al. (2018) illustrate the large uncertainties in methane emission data. Depending on the upscaling methods these authors arrive at global CH4 emission rates from lakes in the range of 78 – 248 Tg CH$_4$ C yr$^{-1}$. Specifically, shallow lakes are known to generally be hot spots in terms of CH$_4$ emissions (Cole et al. 2007; Davidson et al. 2018).**

**Rivers emit around 26.8 Tg CH$_4$ yr$^{-1}$ excluding ebullition (Stanley et al. 2016), however, due to a lack of global data coverage and consistency their role in both carbon transport and storage is not well constrained (Tranvik et al. 2009). In the anthropogenically modified Danube Delta we refer to the internal connections between the main river reaches and the lakes as channels (Kasprak et al. 2016). In general, there is a need for more detailed assessment of the role of methane emissions from rivers and channels as they have been suggested to be more spatiotemporally variable for CH$_4$ than CO$_2$ (Stanley et al. 2016; Natchimuthu et al. 2017).**

**SC6.** L43. To my knowledge, there is no definition of "channel"; aren't these just running waters that are somehow anthropogenically modified, such as many rivers and streams?

**Response: There are a few reviews on classifying channels and streams (see Rosgen Stream Classification System), and these classifications have been discussed by Kasprak et al., 2016 in the framework of a stream channel classification. The channels accessible by boat in the Danube Delta are modified to keep them open for boat access. Overall, we have gone with channels, as where we measured, they are flowing between regions (e.g. between two lakes, or a river to a lake), and some of which are manmade. We have now added a reference in the modified section under line 41.**

**SC7.** L47. "end of line respiration process" sounds like colloquial language, please revise.

**Response: Will be revised, see below Line 48-50**

**SC8.** L48-50. No need to go into pathways of methane production, since it is not at all part of this paper. Also Figure 1 is not really needed, since this study is not about establishing a lake methane budget.

**Response: We will remove Figure 1 and re-focus the paragraph on the transport pathways: "Methane is produced in anaerobic environments, mostly within sediments (Schubert & Wehrli, 2019). Transport mechanisms to the atmosphere include turbulent diffusion trough the water column followed by diffusive gas-exchange across the water- air interface. Methane oxidizing bacteria in the water column reduces methane concentrations depending on the mixing regime (Mayer et al. 2020). At high supersaturation and low hydrostatic pressure bubbles can form and depending on their size ebullition offers a direct pathway from the sediments to the atmosphere (DelSontro et al., 2015).**

**To be deleted: Typically, $CH_4$ is biogenically produced within anaerobic environments (see Fig. 1 for details), where microbial fermentation of organic matter occurs and is controlled by the interplay between input of organic matter and temperature (Stanley et al. 2016).This is generally the end of the line respiration process, through either hydrogenotrophic methanogenesis where oxidation of $H_2$ using $CO_2$ as a terminal electron acceptor produces $CH_4$ or by acetoclastic methanogenesis via the breakdown of simple 50 substrates or acetate, which is a major pathway within the fresh water systems (Whiticar et al. 1986). Other processes also include bubbles transport via ebullition accounting up to 50% of the total flux in certain systems, and generally contributing far larger than that of diffusive fluxes (van Bergen et al. 2019), or more physical processes such as vertical mixing, lateral transport and ground water inputs (Crawford et al. 2014a; Stanley et al. 2016).**

**SC9.** L60-64. Unclear in how far this is relevant introduction to this paper. Instead, focus the introduction on "spatial and temporal variability" (L64), because that's what this paper is about.

**Response: We already suggested changes to this section in response to the other reviewer.**

**SC10.** L66. Unclear what "monitoring approaches tend to stay within one system" means.

**Response: Often monitoring approaches have prioritized one water type (such as lakes). Will be rephrased as 'Given the complexity of inland water systems, especially wetland complexes, monitoring approaches were often focused on only one water type such as a river reach or a lake.'**

**SC11.** L76. Objective 2 should be rephrased, since you do not address global-scale fluxes in this paper, and you only measured one diel cycle.

**Response: This will be rephrased "… data to explore the importance of a diel cycle on local and regional emission rates"**

**SC12.** Section 2.2. Please describe what distinguishes rivers from channels.

**Response: This will be elaborated on. We classed channels as smaller bodies connecting between regions, with only the two larger branches as rivers, given their size, slightly faster flowing and greater depth.**

**SC13.** Figure 2. This map shows the travelled track (how long was it in total?), but couldn't you also make maps that show the concentrations of $CO_2$ and $CH_4$ along this track? This would be a very intuitive way to visualize the data. Also: the yellow lake complexes were not studied and do not need to be highlighted. And instead of the various denominations (3, 4, b1, b2, i-v), what about writing the respective names onto the map?

**Response: We will be adding figures of the concentrations along the tracks – especially for spatial variability visualisation. This figure will be made clearer with the removal of boxes not needed.**

**SC14.** Figure 2 caption: what does "with only slight variations" mean?

**Response: We will omit this statement in the caption, because it is confusing. Instead we mention the deviation in the cruise track in a new sentence following line 104 as follows "Using the small houseboat, the set-up was fixed, and a thorough transect throughout the delta was carried out with extensive lake transects completed in all three seasons for comparability (Fig 2). Due to blockages in the channel between Lake Puiu and Lake Rosu, the transect had to be changed slightly between seasonal campaigns."**

**SC15.** L99-100- Is this the annual temperature range, or daily? And what makes it extreme?

**Response: This is the annual temperature range. It was meant to illustrate that the delta experiences extreme temperatures (30 + to below freezing): The delta is within the temperate climate system, but experiences a broad annual ranges of air temperature from below freezing to more than 30° C (ICDP 2004).**

**SC16.** L102. "thorough" is subjective, and can be skipped.

**Response: Corrected**

**SC17.** L120-125. I have some serious concerns with the way the gas exchange velocity and flux estimates were treated and discussed. Only the concentrations CH4water and CO2water were measured. The concentrations in air were assumed to be at global average, which is doubtful in such a biologically active wetland area. The gas exchange velocity k was scaled from wind speed (unclear where it was measured) for both lakes and rivers (albeit different exponents were used for lakes and rivers). I assume that the channels were treated as the rivers? My point is that k is highly variable at short time scales, and largely driven by hydrodynamics, which in turn varies with wind speed, but also with water flow, hydromorphology, and thermal structure. These sources of variability will vary in time and space and between types of systems, and are very unlikely to be captured by scaling from wind speed. The authors need to acknowledge that, and add some discussion on the reliability of their estimate of k. Given that apparently a study of greenhouse gas emission from the Danube Delta using floating chambers was published recently (Maier et al.), the authors could use the measured k values from that study for calculation of their own fluxes, or to assess in how far the wind speedscaled k values are congruent with measurements of k. The robust treasure of this study are the highly-resolved and repeated concentration measurements, and it needs to be made clearer that the fluxes reported here are estimates, not measurements. Another important aspect of equation 3: the gas exchange velocity and concentration influence each other. A very high k can quickly empty the water of gases and thus lead to low concentrations, and low k prevents emission and can lead to the build-up of high concentrations. It may therefore very well be possible that the sites where the authors have observed high concentrations, the fluxes may not be high if that site was characterized by very stagnant water (possibly in the "hot spot" channel?), instead concentrations might have been high because k and thus flux was low. A relevant paper on the spatial decoupling of k, concentration and emission is Rocher-Ros et al. 2019, L&O, their Figs. 2 and 3. This aspect should be added to the discussion.

**Response: The first point was expressed by the editor as well, therefore alternative calculations were conducted using differing measurements of air concentrations for CH$_4$, such as the nearest Global Greenhouse Gas Reference Network station Hegyhatsal (HUN), Hungary. This made negligible difference in the overall fluxes due to the extreme water-side supersaturation, and therefore is not considered an issue, but this will be stated clearer. For the second point, the wind was taken from the Gorgova station (roughly middle of the delta, will be included in review). We will definitely be acknowledging this point more so in the review as we are very aware k can vary significantly. Maier et al. (2020) measured in different years to this excursion so direct use of their k value would probably not be accurate. However, comparison between the k values produced from the models will be done and added to the supplementary. It will be made clearer that these are estimates. All of this will be addressed in the discussion and it will be made clearer.**

**SC18.** L126. This statement needs a reference.

**Response: This will be added: Schilder et al. (2013).**

**Results and Discussion:**

**SC19.** I wonder if it would not be helpful to separate the Results from the Discussion, and to present the results step by step (concentrations, maps of concentration, then estimates of emission flux, then an upscaled emission for the entire Delta), to then

Discuss the ensemble of the findings.

**Response: Following the advice of the other reviewer we will edit the results and discussion section to improve clarity and remove redundancies. We feel that separating discussion from results might contradict this effort.**

**SC20.** L132. Consistent, not constant.

**Response: Will be changed.**

**SC21.** L133. Another point of serious concern: This study completely lacks statistical testing of the reported differences, and the term "significant" should only be used if a statistical test can support that the difference between e.g. systems or sampling campaigns was statistically significant. The authors must include statistical testing in their revision.

**Response: Correct, also in response to the other reviewer will replace the word "significant", if it is not backed up by statistical analysis. We will use a Kruskal-Wallis test to compare rivers, channels and lakes.**

**SC22.** L134. Using maximum values is not very helpful, better to report means, medians or ranges.

**Response: As the data usually are not normally distributed, we will use medians and ranges.**

**SC23.** L137. What is meant by "water type boundaries"?

**Response: Water type boundaries means areas such as channels leading into lakes, the mixing areas, as stated within the manuscript. This description will be added in the description describing the sectioning processes.**

**SC24.** L138. If I remember right, Crawford et al. studied streams, not channels.

**Response: Yes. Will be corrected.**

**SC25.** L139. "were found to have higher concentrations" – where is this visible?

**Response: This will be implemented visually with the figures stated above. It is visible closer to the edges of the lakes where the wetland is situated.**

**SC26.** L140. What is a "boundary crossover"?

**Response: Between two boundaries, such as where channels enter lakes. Rephrased to 'These boundary crossovers, where higher concentrations were visible to proceeding regions, were due to. . .'.**

**SC27.** L142. On the map, there is channel north of Lake Puiu?

**Response: All areas and locations will be made clearer on the figure, larger maps will also be included into the supplementary .**

**SC28.** L143. Using the term "hot spot anomaly" requires some quantitative and statistical underpinning. It seems from Table 1 that this site was only showing elevated CH4 in Aug, but not in May and Oct. So is this site really significantly different from other sites, i.e. other river reaches, or other lakes, or other channels? Statistical testing is warranted.

**Response: A statistical analysis will be added to this section.**

**SC29.** L146. Briefly explain that in October, macrophytes senesce and can be expected to start decomposing in the water.

**Response: Rephrased: 'The highest median was observed during Oct for rivers, lakes and channels (median: 559, 693 and 1500 nmol L$^{-1}$ respectively), potentially due to macrophyte senesces and decomposition.**

**SC30.** L147. "measurements were not distributed proportionally" – this not only applies to O2 measurements, but to all measurements, so this would affect all your data and conclusions?

**Response: This should not affect the diel cycles, and this would have no effect. This could potentially skew the overall median values; however, this is why we split the data into different categories (lakes, rivers and channels) and attempted to get mapping coverage of each of these regions. Therefore, when analyzing the data while in each separate region, it shouldn't affect the outcomes.**

**SC31.** L150. This sentence is speculative and should be removed.

**Response: We disagree. It is well known that water that spent time in close contact to wetland vegetation looses oxygen and builds up $CO_2$ concentrations (see for instance Zurbrügg et al. 2012). We will rephrase this as: "These values included the 'hot spot'. Wetland waters entering the fluvial systems are often de-oxygenated (Zuidgeest et al. 2016). As this station represents sites receiving water from the wetland it is likely not the only such site in the delta."**

**SC32.** Figure 3. I suggest to present only concentrations, and give some aggregated numbers for fluxes later. Fluxes are only calculated estimates, which are derived from your actual measurements. Instead, also include CO2 concentrations here. And please include statistical testing to infer any differences between categories. Also, please label the panels of this figure. Two observations: O2 saturation was frequently very low, indicating strong respiration in the water or the reed belt. And the distribution of CH4 was very skewed, with generally rather low values, but quite a bunch of very high values.

**Response: Thank you for the constructive comments to improve Figure 3. Concentrations and fluxes will be separated. Panels will be labelled, and statistics between the regions will be inserted. $CO_2$ will be removed from the manuscript as previously stated.**

**SC33.** L152. Fluxes correspond to concentrations because your k estimate is essentially a constant, which k certainly is not in nature. On the contrary, it can be very variable at short scale of space and time. This observation is an artefact.

**Response: This is a very valid comment but not completely true. Fluxes would have been much more accurate if they were based on in situ measured wind measurements. However, we did use wind data from Gorgova (roughly in the middle of the delta) to calculate k and therefore fluxes include the effect of wind speed. This can be observed in Fig. 3 in lakes and channels especially and the k values did vary. But this will be commented on in the manuscript and made clearer.**

**SC34.** L154. For upscaling, it is very important to detail how the calculations were performed, and which assumptions were made, step by step. This was not really the case here.

**Response: As also pointed out in the reply to reviewer 1, we now cite the average values for the three systems in Table 1 on which the upscaling is based.**

**SC35.** L159 "this estimate " – which estimate?

**Response: Meaning the calculated fluxes: „ However, these calculated fluxes include only diffusive emissions."**

**SC36.** L161-164. Confusing that both a 277% and 70% underestimation of total flux are cited. Using the 70% estimate seems more realistic, because that stems from the same system.

**Response: We will delete the reference to the 277% present and focus on the local comparison. The paragraph will start with "In their study over two years, however, Maier et. al., (2020) found evidence that bubble emission of methane in the Danube Delta lakes and channels, potentially accounted for 70%."**

**SC37.** L169-170. Use those floating chamber measurements to calculate k values, which you then can use for your upscaling. It would also be informative to compare the floating chamber measurements of emission to your calculations of emission.

**Response: We will focus on the comparison of the two emission data, the direct flux chamber measurements by Maier et al. (2020) and our calculations.**

**SC38.** Table 1. These are descriptive statistics. Also, please include $CO_2$ here and save $CH_4$ flux for later. "stinky channel hot spot" does not seem appropriate terminology, and it does also not seem to have extremely high concentrations compared to the other channels. And what does the footnote ** mean?

**Response: Terminology will be changed and clarified. When taking the median, the hot spot is not exceptional, except for Aug, however the entire concentration range is far higher. The ** footnote means there was influence on the edges from the channels into the lakes, across the border but meaning they had a few meters of extreme concentrations.**

**SC39.** L173. Please show these correlations, or give regression statistics in the text. This sentence could also be interpreted as an indication that scaling k from wind speed at some met station was not really relevant.

**Response: In response to similar comments by reviewer 1 we will delete this paragraph and discuss patterns and their possible drivers later. Discussed on now L140.**

**SC40.** L175-176. Which external factors? Aren't the most important factors biological and physical?

**Response: We will remove this paragraph see response to 173.**

**SC41.** L179. Change with respect to what? And again, "significantly" requires some form of statistical testing.

**Response: The statement was not meant in a statistical sense and the paragraph will start with "Different processes influence the seasonal carbon turnover and methane production in the delta".**

**SC42.** L183. Concentrations and stuartion of what?

**Response: CH4, this will be implemented!**

**SC43.** L188. I would expect that macrophyte degradation should also be high in the channels, not only the lakes?

**Response: Yes, this will be implemented. "... the process of macrophyte degradation within the delta in both, lakes and channels, was linked ..."**

**SC44.** L189. This could be explored further. With your data, you could make maps and actually at which locations concentration were elevated. For the lakes, you might want to make a correlation between distance from shore and concentration.

**Response: This is a helpful suggestion. We will add more plots illustrating the spatial patterns of CH$_4$ concentration, part of them will be presented as Supplementary information.**

**SC45.** L194. Methanogenesis takes places in anoxic sediments, and I assume the channels don't have very much sediment accumulation at their bottoms?

**Response: Channels have quite high sedimentation rates and the larger ones are dredged to maintain navigation.**

**SC46.** Section 3.1.3. Again, this needs to statistically supported.

**Response: This will be implemented. We will provide a statistical comparison of the hot-spot to the other sites. As also mentioned in the remarks to review 1, we will replace the term "significant influence" by "strong influence"**

**SC47.** L209. Movement of water?

**Response: Yes, will be clarified.**

**SC48.** Section 3.1.4. Unclear what "fluvial" is. Everything minus lakes? Are channels included? And aren't the lakes part of the fluvial delta?

**Response: This is stated on line 212 – rivers and channels.**

**SC49.** L213. "Little evidence" – please show the evidence that you have.

**Response: This will be clarified as follows: "Based on continuous conductivity measurements, we found no evidence for saltwater intrusions from the Black Sea that could suppress methane production by high sulfate concentrations as suggested before (Durisch-Kaiser et al. 2008...)"**

**SC50.** L222. This is expected, since there is very little sediment accumulation expected at the bottom of rivers.

**Response: For sure, and the rivers are far deeper and faster flowing. This was a statement to make it clear.**

**SC51.** L230. Unclear what time period this estimate covers. The three months of measurement? Or the entire year, based on the 3 sampling occasions.

**Response: This is over the 3 campaigns, it will be made clearer: "Overall our calculated mean flux for all months of the three campaigns..."**

**SC52.** L242. This is evidence that the emission might not have changed much, but for as sessing eutrophication, you would need data on phosphorus, nitrogen or chlorophyll.

**Response: Yes, will remove "eutrophication" from the statement.**

**SC53.** L246-249. Here's several statements that require to be supported by showing evidence: enhanced CH4 production, increasing concentrations coming to the lakes, oxidation, visible on the edges of lakes.

**Response: Evidence will be implemented, this will be combined with the spatial plots which was stated before.**

**SC54.** L250-251. Which changes in morphology? What evidence is there for higher productivity in the channels. And what is meant by "macrophyte distributions"?

**Response: The statement is rephrased below and will be illustrated with additional concentration maps. Macrophyte distributions with the lakes were visible from the $O_2$ concentrations, and were shown to change over the lake over the seasons. "The spatial differences and seasonal changes in the surface methane concentrations were far clearer in the lakes than the channels. Distribution of macrophytes in lakes could be linked to the map of $O_2$ and decaying plant biomass explained the high $CH_4$ levels in October."**

**SC55.** L256. Ebullition is also a flux from the sediment.

**Response: This is true, however, diffusive transport and ebullition are usually discussed separately and we will write: "Diffusive release from sediments is usually the primary source of methane in surface waters (Peeters et al. 2019). Ebullition, however, adds a second pathway of $CH_4$ emissions to the atmosphere which is much more variable between systems and locations (see Bastviken . . . '**

**SC56.** L260. With your data, you are in a very good position to explore local dynamics, by making maps and showing them.

**Response: As mentioned, we will expand the discussion of local dynamics and spatial variability and support this with additional maps and improved Figures.**

**SC57.** Section 3.2. This section is far too long, mainly because there was only one true diel cycle measured; during the other diel-cycle measurements, the boat was moving, and thus spatial variability is included in the measurement. Also, the authors lack data that help to explain the diel cycle, e.g. water column profiles of temperature (to address convection) or of gases, measurements of k, or similar. Therefore, the discussion is quite vague. Based on so little data and machnistic understanding, it does not seem warranted to draw the conclusion that diel cycles are important in the Danube Delta, and need to be accounted for (e.g. in the abstract, or L339-341)

**Response: Reviewer 1 remarked that data on the diel cycle are interesting but mentioned that generalizations would take the discussion too far. However, the parallel measurements of temperature and dissolved gases do allow some careful mechanistic interpretation. The movement during the diel cycle was taken into account (line 270), however the spatial variability follows clearly the overall diel pattern, especially as we measured stationary until after sunrise where we saw a decrease in $CH_4$ concentrations.**

**SC58.** Figure 4. I would prefer simpler plots, with time of the x axis and the analytes on the y axis.

**Response: An example for CH4 is shown in Fig. 5. We will add similar plots for the other parameters.**

**SC59.** Figure 5. I would like to see more of this! More maps with concentrations, and further analyses of spatial patterns of elevated (and low) concentrations.

**Response: Good suggestion, as mentioned already, we will add more maps and improve the figures.**

**SC60.** L370. Is there any data or other evidence for high concentrations in the reed bed?

**Response: No measurements were taken in the reed beds because these sites were not accessible by houseboat. We see the influence from the edges of the lakes however, and in Maier et al., 2020 they state evidence for this flowing in from**

the edges too.

---

## Referee Report (RR1)

**General comments**

Authors significantly improved the manuscript. It is an important contribution into methane dynamics in highly heterogenous system of river delta and thorough modification of manuscript by authors allowed to emphasize main assets of this study. I strongly recommend it to be published as it adds important piece into the knowledge about spatial and temporal variability of CH4 in delta system. However there are few minor points, which still needs to be addressed before the publication.

**Specific Comments**

Abstract
Overall, the abstract reads well.

Specific comment:
We found large to extreme diel cycles: The "extreme cycles" doesn't sound correct: What is an extreme diel cycle? Very variable? Please specify

**Introduction:**

Introduction was shortened. Selected parts which distracted the reader from the main path of the manuscript were removed.
Overall it reads smoothly and sets a good background and motivation for this study. Authors significantly improved this part.

Specific comments:
Line 30: Please reformulate part of the sentence: "Due to their significant source strength…"It doesn't read well in English
Line 40-41: Sentence "In the anthropogenically…." Fits more to the method section than to the introduction section. I would suggest to remove it from introduction.
Line 62-73: Authors nicely emphasized the importance of this study, good job!

**Methods:**

The method section is clear and together with references and implemented improvements provides sufficient information about studied area and methods used.
The figure (Fig. 1 ) which describes the studied locations with distinction into channels, river and lakes provides clear overview of the investigated area.

Also, additional information, including streams velocity or information about quasi-stagnant waters, which occur in the delta, provide sufficient justification for implementation of Cole and Caraco gas transfer model to obtain CH4 flux estimates.

**Results and discussion:**

Many points have been clarified, redundant information has been removed and multiple sentences have been rewritten. This allows the reader to follow this section more easily. Result and discussion section reads well, however several parts still need to be addressed.
Specific comments:

Figure 4 provides informative overview of CH4 concentrations in Delta system

Line 151: letter "t" is missing in the word "Throughout"

Line 169-180: This section has been clarified regarding the extrapolation, it is clear now how authors performed the extrapolation.

Line 176: word "estimated" is missing letter "e"

Line 224-227: Despite the attempt made by authors to clarify this sentence, it still needs to be rephrased as it is difficult to grasp the main message of this sentence.

Line 236: word "were" is missing letter "e"

Line 251: Please change from 'hot spot' measured the largest concentrations" to "the largest concentration was measured….."

Line 258: Please rephrase: Oct by itself cant have median and percentiles, fluxes can.

Line 264: The sentence seems not complete: Higher median than where?

Line 264: Are you sure about this number (2030) ?

Line 287: "as the CH4 due to being quickly oxidized" : This sounds like authors actually measured CH4 oxidation (which was not the case). Please rephrase

Line 307: Please indicate correct letters for the fig.7

Line 307-315: Please use past tense while reporting results, for example "diel cycle showed" instead of "diel cycle shows". It is important to be coherent with other parts of manuscript where past tense was used to describe the results.

Line 311: (Fig9): Numbering of the figures needs to follow description in the text. Thus, if this Fig appears in the text for the first time at this point, it should be Fig. 8, not Fig.9. Please change accordingly in following parts of the manuscript.

Line 327: Please change "is" to "was"

Line 330-343: Please use past tense while reporting results to be coherent with other parts of manuscript

Line 349: Fig. 8: Nice visualization! However, where is this fig discussed in the text? Please clarify

Line 396-397: This sentence sounds like finding of the study (which I don't think it is). Please rephrase

---

## Author Response (AR2)

**General comments**

Authors significantly improved the manuscript. It is an important contribution into methane dynamics in highly heterogenous system of river delta and thorough modification of manuscript by authors allowed to emphasize main assets of this study. I strongly recommend it to be published as it adds important piece into the knowledge about spatial and temporal variability of CH4 in delta system. However there are few minor points, which still needs to be addressed before the publication.

**Specific Comments**

Abstract
Overall, the abstract reads well.

Specific comment:
We found large to extreme diel cycles: The "extreme cycles" doesn't sound correct: What is an extreme diel cycle? Very variable? Please specify

*Response: Correct, a diel cycle that has a large gradient change overnight. In our case a diel cycle that has a great change in CH4 concentration overnight compared to that of the starting concentration.*

**Introduction:**

Introduction was shortened. Selected parts which distracted the reader from the main path of the manuscript were removed.
Overall it reads smoothly and sets a good background and motivation for this study. Authors significantly improved this part.

*Response: Thank you!*

Specific comments:
Line 30: Please reformulate part of the sentence: "Due to their significant source strength…"It doesn't read well in English

*Response: Changed from 'Due to their significant CH4 source strength, inland waters have seen an increase in attention (…' to 'Inland waters are known to have a significant CH4 source strength and therefore, have seen an increase in attention (…'*

Line 40-41: Sentence "In the anthropogenically…." Fits more to the method section than to the introduction section. I would suggest to remove it from introduction.

*Response: This has been moved.*

Line 62-73: Authors nicely emphasized the importance of this study, good job!
*Response: Thank you!*

**Methods:**

The method section is clear and together with references and implemented improvements provides sufficient information about studied area and methods used.
The figure (Fig. 1 ) which describes the studied locations with distinction into channels, river and lakes

provides clear overview of the investigated area.

Also, additional information, including streams velocity or information about quasi-stagnant waters, which occur in the delta, provide sufficient justification for implementation of Cole and Caraco gas transfer model to obtain CH4 flux estimates.

**Results and discussion:**
Many points have been clarified, redundant information has been removed and multiple sentences have been rewritten. This allows the reader to follow this section more easily. Result and discussion section reads well, however several parts still need to be addressed.
Specific comments:

Figure 4 provides informative overview of CH4 concentrations in Delta system

Line 151: letter "t" is missing in the word "Throughout"

***Response: This has been implemented***

Line 169-180: This section has been clarified regarding the extrapolation, it is clear now how authors performed the extrapolation.

Line 176: word "estimated" is missing letter "e"

***Response: This has been implemented***

Line 224-227: Despite the attempt made by authors to clarify this sentence, it still needs to be rephrased as it is difficult to grasp the main message of this sentence.

***Response: Changed from 'Given the dramatic change within the concentrations and properties of the water, such as the water temperature decreasing the further away from the channel we travelled into the 'hot spot', even within summer, this would further provide evidence from cooler groundwaters or potential waters from the reed beds also suggested by Maier et al. (2021).' To 'Given the dramatic change within the concentrations and properties of the water, i.e. water temperature decreasing the further inwards we travelled, this would further provide evidence of influence from cooler groundwaters or potential waters from the reed beds also suggested by Maier et al. (2021).***

Line 236: word "were" is missing letter "e"

***Response: This has been implemented***

Line 251: Please change from 'hot spot' measured the largest concentrations" to "the largest concentration was measured....."

***Response: This has been implemented***

Line 258: Please rephrase: Oct by itself cant have median and percentiles, fluxes can.

***Response: Changed from 'Comparing Oct to May and Aug for rivers, it had the largest percentile range and median' to 'Comparing fluxes from Oct to May and Aug fluxes for rivers, it had the largest percentile range and median'***

Line 264: The sentence seems not complete: Higher median than where?

***Response: Changed from 'Overall our calculated mean flux for all months of the three campaigns from the fluvial delta was 594 ± 525 umol m-2 h-1, within the diffusive mean from the overall literature (342.5 ± 1062.5 umol m-2 h-1; Sanley et al., 2016). However, we found a far higher median of 473 umol m-2 h-1 (compared to 33.3 umol m-2 h-1).' To 'Overall our calculated mean flux for all months of the three campaigns from the fluvial delta was 594 ± 525 umol m-2 h-1, within the diffusive mean from the overall literature (342.5 ± 1062.5 umol m-2 h-1; Sanley et al., 2016), yet with a far higher median of 473 umol m-2 h-1 (compared to 33.3 umol m-2 h-1).'***

Line 264: Are you sure about this number (2030) ?

***Response: Rounded to 3 figures, yes***

Line 287: "as the CH4 due to being quickly oxidized" : This sounds like authors actually measured CH4 oxidation (which was not the case). Please rephrase

*Response: Changed from 'This inflow was only visible on the edges of the lakes and although had influence on the overall concentration, were seen as outliers as the CH4 due to being quickly oxidized (Fig. 6)' to 'This inflow was only visible on the edges of the lakes and although had influence on the overall concentration, were seen as outliers as the CH4 appeared to potentially be quickly oxidized (Fig. 6)'*

Line 307: Please indicate correct letters for the fig.7
*Response: This has been implemented*

Line 307-315: Please use past tense while reporting results, for example "diel cycle showed" instead of "diel cycle shows". It is important to be coherent with other parts of manuscript where past tense was used to describe the results.
*Response: This has been implemented throughout*

Line 311: (Fig9): Numbering of the figures needs to follow description in the text. Thus, if this Fig appears in the text for the first time at this point, it should be Fig. 8, not Fig.9. Please change accordingly in following parts of the manuscript.
*Response: Figures have been re-arranged*

Line 327: Please change "is" to "was"
*Response: This has been implemented*

Line 330-343: Please use past tense while reporting results to be coherent with other parts of manuscript
*Response: This has been implemented and checked throughout*

Line 349: Fig. 8: Nice visualization! However, where is this fig discussed in the text? Please clarify
*Response: Thank you and this has been implemented*

Line 396-397: This sentence sounds like finding of the study (which I don't think it is). Please rephrase
*Response: Changed from 'The diel cycle within the lake was consistent with stratification over the day, where vast amounts of organic carbon from macrophytes created anoxic subsurface waters, which slowly and steadily mixed during the night' to 'The diel cycle within the lake was consistent with the potential stratification over the day, where potentially vast amounts of organic carbon from macrophytes created anoxic subsurface waters, which slowly and steadily mixed during the night'.*